# Chemokines cooperate with TNF to provide protective anti-viral immunity and to enhance inflammation

Alí Alejo[1], M. Begoña Ruiz-Argüello[1,4], Sergio M. Pontejo [2,5], María del Mar Fernández de Marco[2,6], Margarida Saraiva[3,7], Bruno Hernáez[2] & Antonio Alcamí[2,3]

The role of cytokines and chemokines in anti-viral defense has been demonstrated, but their relative contribution to protective anti-viral responses in vivo is not fully understood. Cytokine response modifier D (CrmD) is a secreted receptor for TNF and lymphotoxin containing the smallpox virus-encoded chemokine receptor (SECRET) domain and is expressed by ectromelia virus, the causative agent of the smallpox-like disease mousepox. Here we show that CrmD is an essential virulence factor that controls natural killer cell activation and allows progression of fatal mousepox, and demonstrate that both SECRET and TNF binding domains are required for full CrmD activity. Vaccination with recombinant CrmD protects animals from lethal mousepox. These results indicate that a specific set of chemokines enhance the inflammatory and protective anti-viral responses mediated by TNF and lymphotoxin, and illustrate how viruses optimize anti-TNF strategies with the addition of a chemokine binding domain as soluble decoy receptors.

[1] Centro de Investigación en Sanidad Animal; Instituto Nacional de Investigación y Tecnología Agraria y Alimentaria, Valdeolmos, Madrid 28130, Spain. [2] Centro de Biología Molecular Severo Ochoa (Consejo Superior de Investigaciones Científicas and Universidad Autónoma de Madrid), Cantoblanco, Madrid 28049, Spain. [3] Department of Medicine, University of Cambridge, Addenbrooke's Hospital, Cambridge CB2 2QQ, United Kingdom. [4] Present address: Progenika Biopharma, 48160, Derio, Spain. [5] Present address: National Institutes of Health, Bethesda, Maryland 20892, USA. [6] Present address: Animal & Plant Health Agency, Addlestone, Surrey KT15 3NB, UK. [7] Present address: Institute for Molecular and Cell Biology, 4200-135 Porto, Portugal. These authors contributed equally: Alí Alejo, M. Begoña Ruiz-Argüello. Correspondence and requests for materials should be addressed to A.Aí. (email: aalcami@cbm.csic.es)

A unique immune evasion mechanism employed by poxviruses and herpesviruses is the production of soluble binding proteins or secreted versions of host receptors that neutralize cytokines[1–4]. The poxvirus homologs of host tumour necrosis factor TNF (TNF) receptors (vTNFRs) block the pro-inflammatory activity of some TNF superfamily (TNFSF) ligands. Five vTNFRs have been described in poxviruses, a viral homolog of host TNFSF receptor CD30 and four TNF inhibitors named cytokine response modifiers B, C, D, and E (CrmB, CrmC, CrmD, and CrmE). vTNFRs are differently expressed among viral species and show distinct binding and inhibitory properties[5–13]. While CrmE and CrmC are specific TNF inhibitors, CrmD and CrmB block TNF and lymphotoxin α (LTα). Furthermore, we have recently described that vTNFRs can inhibit membrane TNF and that CrmD and CrmB neutralize another TNFSF ligand, LTβ[14,15]. In addition to the cysteine-rich domains (CRDs), characteristic of the ligand binding region of cellular TNFRs, CrmB and CrmD have a C-terminal domain unrelated to host proteins that binds chemokines and was named SECRET (smallpox virus-encoded chemokine receptor) domain[16]. The crystal structure of the CrmD SECRET domain

showed a beta-sandwich fold similar to that of the viral chemokine binding proteins (vCKBPs) 35-kDa and A41, but a different chemokine interaction region may confer its distinct narrower binding specificity[17–19]. Such variety of activities may provide poxviruses the ability to differentially block chemokines involved in distinct anti-viral responses, to inhibit chemokines at different stages of infection in the host or to simultaneously inhibit chemokines and TNF. Interestingly, the beta-sandwich fold of vCKBPs has also been observed in other unrelated poxviral proteins including CPXV203, a major histocompatibility complex I binding protein encoded by cowpox virus (CPXV), and GIF, the granulocyte-macrophage colony-stimulating factor and interleukin 2 inhibitor of the parapox Orf virus[4,20]. To reflect such diverse range of immunomodulatory activities this folding has been named as poxvirus immune evasion domain[4,20].

ECTV is a mouse-specific orthopoxvirus[21,22] genetically related to vaccinia virus (VACV), variola virus (VARV) (the causative agent of human smallpox) and monkeypox virus (MPXV)[23,24], a human pathogen whose incidence is increasing due to the cessation of mass smallpox vaccination in Africa[25,26]. Susceptible

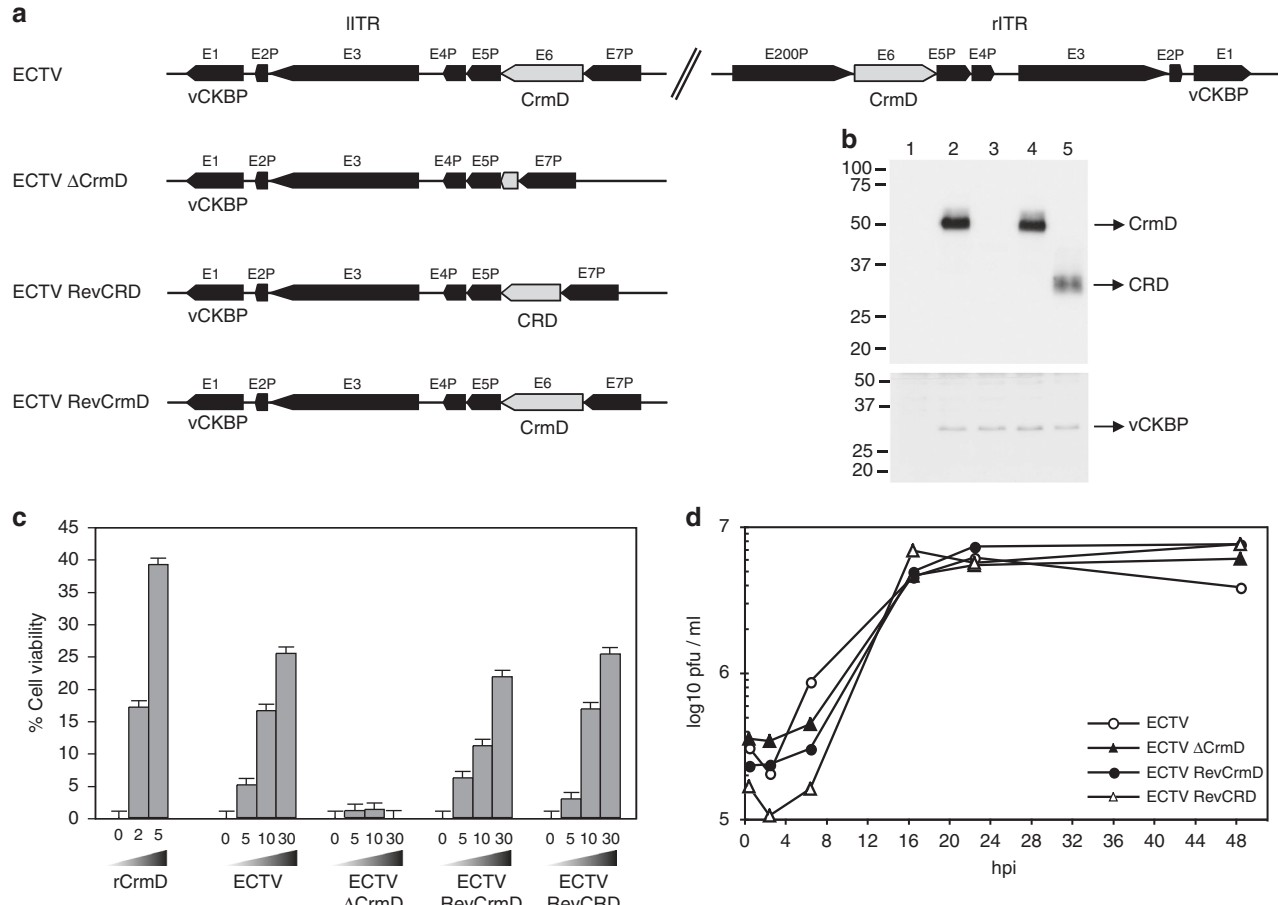

**Fig. 1** ECTV CrmD recombinant viruses. **a** Schematic diagram of the genomic structure of the recombinant ECTVs generated for the study of the role of ECTV CrmD in mousepox pathogenesis. The names of the genes flanking the *CrmD* locus are indicated. In gray, the *CrmD* locus in the parental and recombinant viruses is shown. Both left inverted terminal repeat (lITR) and right inverted terminal repeat (rITR) are represented for ECTV, whereas the lITR is shown for the other viruses. **b** Western blot analyses using anti-CrmD and anti-35-kDa vCKBP antisera of supernatants from BSC-1 cells that were mock-infected (1) or infected with ECTV (2), ECTVΔCrmD (3), ECTVRevCrmD (4) or ECTVRevCRD (5) at a multiplicity of infection of 5 PFU/cell and harvested at 24 h post-infection. The position of the respective proteins is indicated by arrows. Molecular size markers in kDa are shown on the left. **c** TNF-induced cytotoxicity assay. Increasing amounts of recombinant ECTV CrmD (rCrmD, in μg/ml) or supernatants (equivalent to 5, 10 or 30 × 10³ cells) obtained as in **b** were added to block the effect of TNF on L929 cells. The values obtained in the presence of mock-infected cell supernatants have been substracted in each case. Data are mean +/− standard error of the mean (SEM) of triplicate samples. **d** Single-step growth curves of the indicated viruses. BSC-1 cells were infected with 5 PFU/cell and the virus production was titrated at the indicated hour post-infection. The means of duplicate samples are shown

**Table 1 Mortality rate determination of CrmD recombinant ECTVs in susceptible BALB/c mice**

| Dose (pfu) | ECTV | | ECTV RevCrmD | | ECTV ΔCrmD | | ECTV RevCRD | |
|---|---|---|---|---|---|---|---|---|
| | Survivors | MTD | Survivors | MTD | Survivors | MTD | Survivors | MTD |
| 10 | 4/10 | 10.0 (9–11) | 0/10 | 11.6 (9–21) | 10/10 | n.a. | 10/10 | n.a. |
| 10e2 | 0/5 | 11.8 (9–18) | 1/5 | 12.8 (10–11) | 5/5 | n.a. | 5/5 | n.a. |
| 10e3 | 0/5 | 10.4 (9–14) | 0/5 | 10.2 (9–13) | 5/5 | n.a. | 4/5 | 10 |
| 10e4 | n.d. | n.d. | 0/5 | 8.6 (8–10) | 5/5 | n.a. | 5/5 | n.a. |
| 10e5 | n.d. | n.d. | 0/5 | 9.6 (8–13) | 5/5 | n.a. | 3/5 | 9 (8–10) |
| 10e6 | n.d. | n.d. | n.d. | n.d. | 5/5 | n.a. | n.d. | n.d. |
| 10e7 | n.d. | n.d. | n.d. | n.d. | 4/5 | 14 | 1/5 | 12.5 (10–16) |

Groups of 5 or 10 BALB/c female mice were infected s.c. in the left hind footpad with different doses of the indicated viruses. The number of survivors at 32 dpi and the mean time to death (MTD) and survival range (in parenthesis) in days for each condition are shown
*n.a.* not applicable, *n.d.* not determined

strains of mice infected with ECTV develop mousepox, a severe disease that constitutes a good model for smallpox. ECTV infection of susceptible mouse strains via the s.c. route has been exploited as a model of generalized virus infections, genetic resistance to disease, and viral immunology[21,22,27]. In ECTV, CrmD is the only secreted TNFR. Similarly, both VARV and MPXV express a single vTNFR with similar characteristics, CrmB. By contrast, CPXV expresses four distinct vTNFRs[13]. In addition, ECTV and other poxviruses encode intracellular proteins that inhibit TNF-induced signalling, underscoring the importance of TNF in antiviral reponses[18,28].

However, our knowledge of the role of TNF and LT in the control of poxviral infections in vivo is limited. Knockout mice lacking both TNFR1 and TNFR2 showed a slightly increased susceptibility to ECTV and elevated viral replication, with 60% of the infected animals succumbing to mousepox while all WT mice resisted infection[29]. A direct antiviral activity of TNF has been proposed using a recombinant VACV expressing TNF[30]. This direct effect was substantiated in TNF-deficient mice infected with VACV, which showed a modest (two-fold) reduction in LD50 as compared to WT mice, that was accompanied by an increased virus load but not by a diminished T cell response[31]. Although resistance to mousepox was associated with Th1-like cytokine expression, including TNF, blockade of TNF using monoclonal antibodies did not affect the generation of NK cell and CTL responses, virus clearance or resistance to ECTV infection[32] and treatment with TNF did only reduce the mortality rate from 100% to 70% in susceptible BALB/c mice[33]. VACV-infected TNFR2-deficient C57BL6 mice produced higher viral titers in spleens and livers and reduced numbers of inflammatory cell foci in the liver, as compared to WT mice[34].

A contribution of vTNFRs to pathogenesis was initially shown with a CPXV lacking CrmB, but expressing other vTNFRs, which displayed an increased LD50 in infected mice after intracranial inoculation, a route of infection not natural for poxviruses[35]. Inactivation of a CrmB homolog (M-T2) from myxoma virus reduced clinical signs of illnesss in infeced rabbits[36]. However, the reported attenuation in the initial studies cannot be formally atributed solely to the absence of the vTNFR since the selection of inadvertent mutations elsewhere in the viral genome was not controlled with the construction of revertant viruses or by sequencing the complete viral genome. Additional studies with the VACV vaccine strain USSR showed that deletion of CrmC or CrmE caused no effect in virulence or a very mild attenuation not affecting mortality, respectively, after i.n. infection of mice[37]. Recombinant VACV strain Western Reserve expressing CrmC, CrmB or CrmE displayed increased virulence less than 10-fold in an i.n. mouse model, but high virus doses were required to cause disease because the recombinant viruses were deficient in the thymidine kinase gene[37,38]. Definitive studies addressing the role

of vTNFRs in viral pathogenesis using virulent poxviruses in their natural host are lacking.

Here we show that CrmD is an essential virulence factor as deletion of CrmD from ECTV resulted in a dramatic attenuation phenotype, generating an avirulent virus that induced strong NK cell and CD8 T cell responses but did not establish fatal mousepox. This demonstrates a critical role of TNF and a reduced set of chemokines in anti-viral defense. Moreover, this unique model of virus infection in a natural host, together with the construction of mutant viruses, allowed us to dissect the relative contribution of TNF and chemokine activities in vivo. We report that expression of the anti-TNF (CRD domain) or anti-chemokine (SECRET domain) activities of CrmD are not sufficient on their own to confer full virulence to ECTV, suggesting that the function of some chemokines complement TNF in protection against viruses. Furthermore, immunization of mice with recombinant CrmD protected from a lethal ECTV challenge.

## Results

**ECTV CrmD is an essential virulence factor.** To address the role of CrmD in mousepox pathogenesis, we generated recombinant ECTVs in the Naval strain (Fig. 1a). An ECTV CrmD deletion mutant was obtained (ECTVΔCrmD) with both copies of the CrmD gene deleted. As a control for the selection of inadvertent mutations in other genes during the generation of ECTVΔCrmD, a revertant virus (ECTVRevCrmD) with both copies of the full length CrmD gene restored was constructed. To study the differential contribution of TNF- vs chemokine- inhibitory activities of CrmD, a virus expressing only the TNF binding domain of CrmD, composed of CRDs (ECTVRevCRD), was constructed. The complete genome sequence of ECTVRevCRD confirmed the correct incorporation of two copies of the truncated CrmD TNF binding domain and the absence of additional mutations. Replication of recombinant viruses was comparable to that of the parental ECTV (Fig. 1d). As expected, ECTVΔCrmD-infected cell supernatants showed neither expression of CrmD protein nor TNF blocking activity, whereas infections with either parental or revertant viruses showed similar levels of CrmD and TNF inhibitory activity (Fig. 1b, c). ECTVRevCRD-infected cells expressed a truncated CrmD protein that inhibited TNF activity as efficiently as supernatants from ECTV- or ECTVRevCrmD-infected cells. As a control, all samples expressed similar amounts of the secreted 35-kDa vCKBP[39] (Fig. 1b).

Viral virulence was determined in susceptible BALB/c mice s.c. in the footpad (Table 1). Only four out of 20 animals infected with 10 PFU of either parental or ECTVRevCrmD viruses survived the disease, with an estimated LD50 of less than 10 PFU. In contrast, ECTVΔCrmD was severely attenuated, as only one animal out of five died when $10^7$ PFU were administered

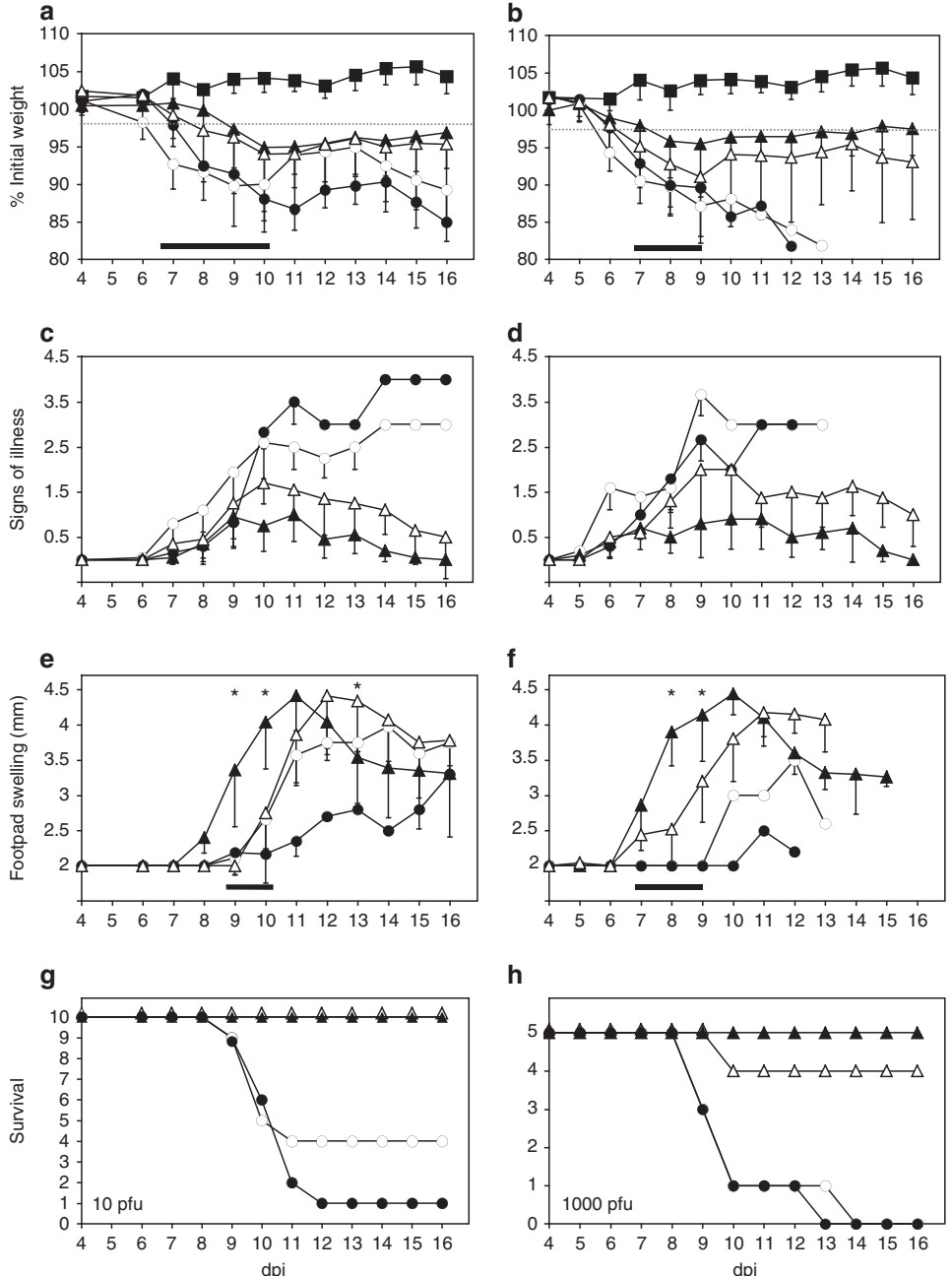

**Fig. 2** Mousepox pathogenesis in the absence of ECTV CrmD. Groups of 10 (**a, c, e, g**) or 5 (**b, d, f, h**) BALB/c mice were inoculated s.c. in the left hind footpad with 10 PFU or 1000 PFU, respectively, of ECTV (white circles), ECTVΔCrmD (black triangles), ECTVRevCrmD (black circles), ECTVRevCRD (white triangles) or PBS alone (black squares), and monitored daily for weight loss (**a, b**), signs of illness (**c, d**) and footpad swelling (**e, f**), as indicated. Data are shown as mean +/− SEM. Statistical analyses were performed using multiple t-tests with false discovery rate at $Q = 1\%$. Analyses were performed up to times post infection at which survival rates in the corresponding group were above 50%. Black bars indicate time points at which significant differences ($p < 0.01$) between ECTV− / ECTVRevCrmD− and ECTVΔCrmD-inoculated mice were found. Asterisks indicate time points at which significant differences ($p < 0.01$) between ECTVΔCrmD and ECTVRevCRD-inoculated mice were found. **g**, **h** The number of surviving animals during the course of infection. Data are from one experiment representative of three independent experiments

while all other mice survived. This difference of at least six orders of magnitude in LD50 of ECTVΔCrmD as compared to ECTV indicated that CrmD is an essential virulence factor in mousepox and that its deletion renders ECTV practically avirulent. Reintroduction of both copies of the full length CrmD into the genome of ECTVΔCrmD restored virulence, demonstrating an exclusive CrmD-mediated effect. Interestingly, the LD50 estimated for mice infected with ECTVRevCRD was around 10^5 PFU, showing attenuation of the virus lacking only the

chemokine inhibitory domain as compared to parental and revertant ECTVRevCrmD viruses. This indicated that the SECRET domain was essential for pathogenesis.

**ECTV CrmD controls an inflammatory reaction in vivo.** Infected animals were monitored daily for weight loss, signs of illness and footpad swelling (Fig. 2). As shown for two viral doses (10 PFU and 1000 PFU per animal), mice infected with

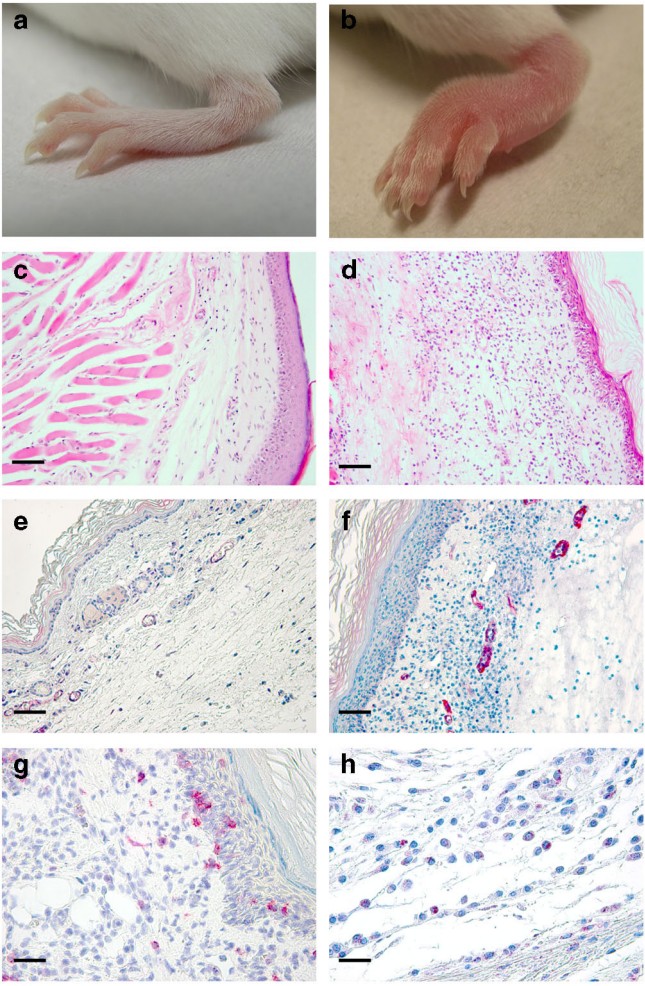

**Fig. 3** Inflammatory response in the footpad of ECTVΔCrmD-infected mice. Left hind foot (**a, b**), H&E staining (**c, d**) and anti-ICAM-1 staining (**e, f**) of zinc-fixed footpad sections of representative BALB/c mice infected with 1000 PFU of ECTV (**a, c, e**) or ECTVΔCrmD (**b, d, f**) at 7 dpi. Anti-CD4 staining (**g**) of zinc-fixed and anti-CCR10 staining (**h**) of formalin-fixed footpad sections of representative ECTVΔCrmD-infected mice are shown. No cell infiltrate was observed in the ECTV-infected tissues. For all immunohistochemistry analyses, positively labelled cells appear in red color. Scale bar, 100 µm (**c–f**) and 25 µm (**g, h**)

ECTVΔCrmD suffered less severe weight loss and signs of illness as compared to mice infected with parental or revertant ECTV-RevCrmD viruses. Weight of ECTVΔCrmD-infected mice did not drop below 95% of their initial value and mean scores peaked at around 1 in a scale ranging from 0 for a healthy individual to 4 for a severely diseased animal. ECTVΔCrmD-infected mice had fully recovered from disease by 16 days post-infection (dpi). Similarly, ECTVRevCRD-infected mice showed reduced weight loss and signs of illness, with surviving mice fully recovering (Fig. 2). In this case, the differences were more apparent at the lower doses. The thickness of the site of virus inoculation (footpad) was assessed as a measure of inflammatory response. In ECTV and ECTVRevCrmD-infected animals only a few individuals responded with footpad swelling, starting at 10 dpi, when most of the mice had succumbed to infection. However, ECTVΔCrmD-infected mice showed a strong footpad swelling starting 2–3 days earlier and peaking by 10–11 dpi at values higher than those observed with parental virus (Fig. 2). Thus, CrmD efficiently controls footpad swelling, consistent with its proposed immunomodulatory role (Fig. 3a, b).

In mice infected with ECTVRevCRD, expressing the TNF but not the chemokine blocking activity of CrmD, footpad swelling was significantly delayed as compared to that observed in the absence of CrmD, showing that TNF activity in vivo is crucial for an inflammatory reaction. However, inflammation did still occur, which could reflect an incomplete blockade of TNF activity as well as the activity of other proinflammatory stimuli, such as the chemokines not blocked due to the absence of the SECRET domain in this virus. Haematoxilin and eosin (H&E) staining of sections of the footpad of ECTVΔCrmD-infected mice showed the presence of a large inflammatory infiltrate with edema in the dermis, which was not detected or much reduced in ECTV-infected mice (Fig. 3c, d). The infiltrate was composed mainly of lymphocytes (Fig. 3g) and macrophages with a few polymorpho-nuclear leukocytes. Immunostaining showed the presence of CD4 $^{+}$ T cells (10% CD3$^{+}$; 12% CD4$^{+}$) and some B cells (5%), but no CD8$^{+}$ cells were detected. Sections were stained for expression of inter cellular adhesion molecule-1 (ICAM-1), an integrin ligand overexpressed on endothelial cells of the postcapillary venules in response to proinflammatory cytokines such as TNF[40]. More than 50% of the vessels in the footpads of ECTVΔCrmD-infected mice showed intense ICAM-1 staining, while only around 25% of the vessels expressed ICAM-1 in ECTV- or ECTVRevCrmD-infected mice (Fig. 3e, f), indicating a role for CrmD in controlling inflammation through TNF inhibition. Our previous studies defined the binding specificity of CrmD for human chemokines[16], and we have characterized here the interaction of CrmD with mouse chemokines (Fig. 4). Consistent with the anti-chemokine activity of CrmD, immunohistochemistry of footpad sections showed that approximately 30% of the infiltrating cells in ECTVΔCrmD-infected mice by 7 dpi expressed the CCR10 chemokine receptor (Fig. 3h) whereas no CCR10-expressing cells were observed in ECTV infections. The CrmD SECRET domain interacts with the mouse chemokines Ccl28 and Ccl27, the latter with higher affinity (Fig. 4), which are recognized by CCR10. This indicates that the SECRET domain may contribute to the inhibition of chemokine-directed cell migration in vivo. Consistently, a similar amount of CCR10-expressing cells was detected in the footpad infiltrate of ECTVRevCRD-infected mice.

**Virus replication is restricted in the absence of ECTV CrmD.** The extent of viral replication and dissemination in the host in the absence of CrmD was analyzed. Viral replication was apparently not hindered at the site of inoculation, as assessed by anti-virus and anti-CrmD staining of sections from footpads of infected mice at 7 dpi (Fig. 5c–h). This also showed that ECTV replicates in vivo in the absence of CrmD and the stability of the truncated CRD protein in vivo. Staining of sections from footpads of uninfected mice showed the specificity of the anti-virus and anti-CrmD antibodies (Supplementary Fig. 1). As shown in Fig. 5a, b, by 3 dpi, all four viruses had reached both the spleen and liver, with no significant differences in the viral titers among them, suggesting that the absence of CrmD did not affect the capacity of the virus to spread to its secondary replication sites. However, both the parental and the revertant ECTVRevCrmD viruses reached high titers by 7 dpi in the spleen (Fig. 5a) and liver (Fig. 5b), whereas ECTVΔCrmD titers were reduced by 2 and 4 log units in these organs, respectively. This shows that ECTV replication in spleen and liver is controlled by the host in the absence of CrmD. The expression of the TNF inhibitory domain of CrmD by ECTVRevCRD fully restored ECTV infectivity in spleen, but not in the liver, suggesting that the relative contribution of TNF activity for protection against virus replication in vivo might be organ-dependent. These results also suggest that chemokine inhibition by the SECRET domain may

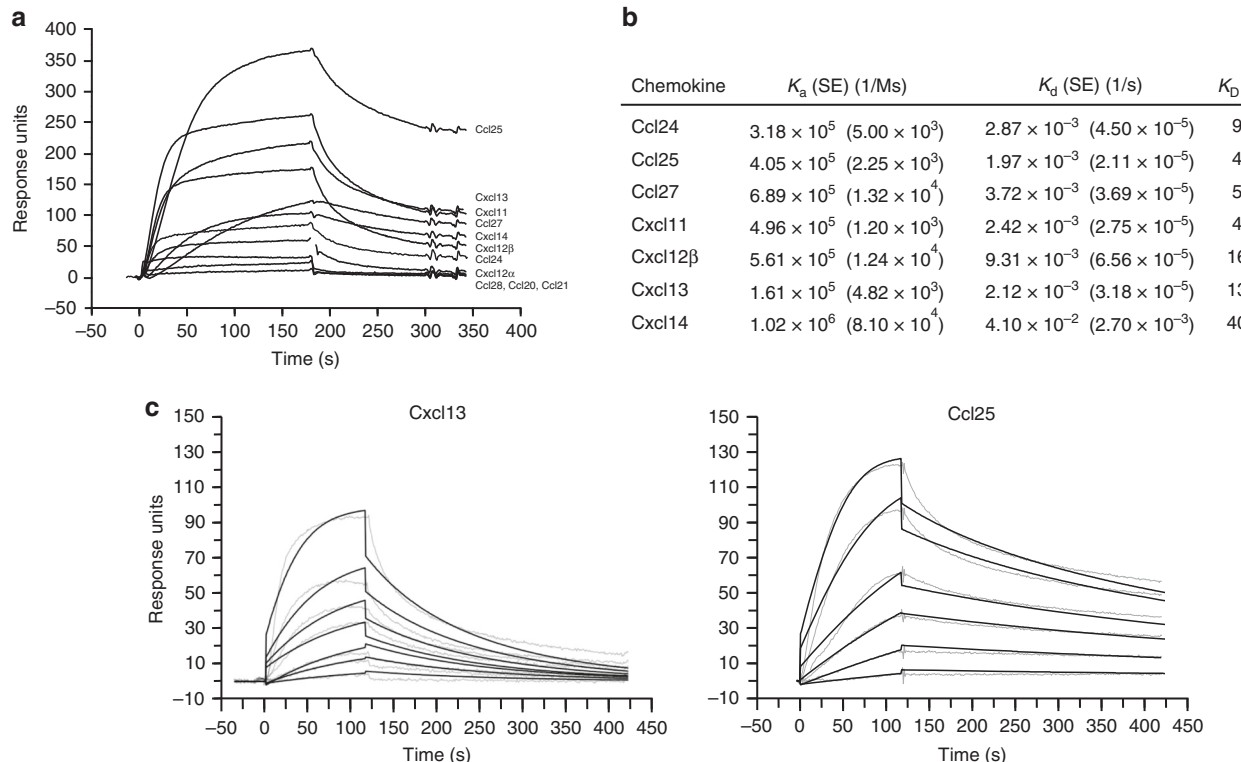

**Fig. 4** CrmD binding affinity for mouse chemokines. **a** Binding screening of mouse chemokines to CrmD. Recombinant CrmD was immobilized on a CM4 SPR biosensor chip at 1500 RU. Chemokines (100 nM) were injected at 10 μl/min flow rate in HBS-EP buffer. Binding was monitored during 180 s followed by a 120 s dissociation periods. **b** Binding kinetic constants $K_a$ (association), $K_d$ (dissociation) and $K_D$ (affinity) and their corresponding SEM for the CrmD interaction with some of its chemokine ligands. For affinity determination purposes, CrmD was immobilized on a CM4 SPR biosensor chip at 900 RUs. The binding of increasing concentrations of chemokine (1–50 nM), injected at a 30 μl/min flow rate, was recorded during 120 s followed by a 300 s dissociation. Binding curves were fitted according to a 1:1 Langmuir model. **c** Two examples of affinity determination fittings are shown

be especially important for virus replication in the liver. All mice infected with either ECTVΔCrmD or ECTVRevCRD survived the infection and virus was being cleared by 11 dpi (Fig. 5a, b). Limited virus replication in the absence of CrmD was accompanied by reduced necrosis of the infected organs (Fig. 5j, m, Table 2, Supplementary Fig. 2), which was also apparent in the liver in the case of ECTVRevCRD-infected mice (Fig. 5n and Table 2). Additionally, an increased inflammation of the liver at 7 or 11 dpi was observed in the absence of CrmD (Fig. 5m and Table 2). Altogether, these results showed that in the absence of CrmD ECTV replication can be controlled by the host and suggest that reduced liver damage is the cause for survival of infected mice.

**ECTV requires inhibition of chemokines and TNF for virulence.** To further assess the role of the CrmD SECRET domain in mousepox pathogenesis, a recombinant virus was constructed in which CrmD was replaced by a CrmD variant bearing a point mutation (N77F) that lacks TNF binding and inhibitory activity while maintaining its chemokine inhibitory activity. Figure 6 shows the binding and biological properties of the purified recombinant CrmD N77F mutant, demonstrating that it has lost the TNF inhibitory activity but retains the ability to inhibit chemokine-mediated cell migration, with a similar potency as that shown by CrmD. The complete genome sequence of this virus, termed ECTVRevSECRET, confirmed the incorporation of two copies of the CrmD N77F mutant gene and that no other mutations that may influence virus virulence were present. ECTVRevSECRET replicated efficiently in cell culture and expressed the mutated protein to similar levels than the

parental virus, considering the loading control of vCKBP (Fig. 7b). As expected, supernatants of ECTVRevSECRET-infected cells did not show TNF inhibitory activity (Fig. 7c).

The virulence of ECTVRevSECRET was assessed in susceptible BALB/c mice infected s.c. in the footpad with virus doses ranging from $10^4$ to $10^6$ PFU per animal. With only one animal out of five succumbing to mousepox after infection with the highest dose tested (Fig. 7d, e), ECTVRevSECRET was nearly as severely attenuated as the ECTVΔCrmD mutant. ECTVRevCRD, expressing the TNF binding domain, was slightly more virulent than ECTVRevSECRET, expressing the chemokine binding activity. ECTVRevSECRET was able to replicate in vivo and to reach the spleen, but replicated to levels lower than ECTVRevCRD and ECTVRevCrmD (Fig. 7f). As shown before, infection with $10^6$ PFU of ECTVΔCrmD produced a strong footpad swelling starting at 5 dpi, that was impaired by expression of CrmD (ECTVRevCrmD) or the TNF binding domain of CrmD (ECTVRevCRD) (Fig. 7e). Expression of the chemokine binding activity of CrmD by ECTVRevSECRET was also able to block this inflammatory reaction, albeit to a reduced degree (Fig. 7e). Altogether, these results showed that the SECRET domain chemokine inhibitory activity per se is not able to act as a virulence factor, suggesting that its role in pathogenesis is only apparent when the TNF inhibitory activity is also expressed by the virus.

**Modulation of NK cell and CD8 T cell responses by CrmD.** It has been shown that upon footpad inoculation of ECTV, the NK cell response during the first 4 dpi in the draining popliteal LN (DPLN) and the T cell response at late stages (peak at 7 dpi) in

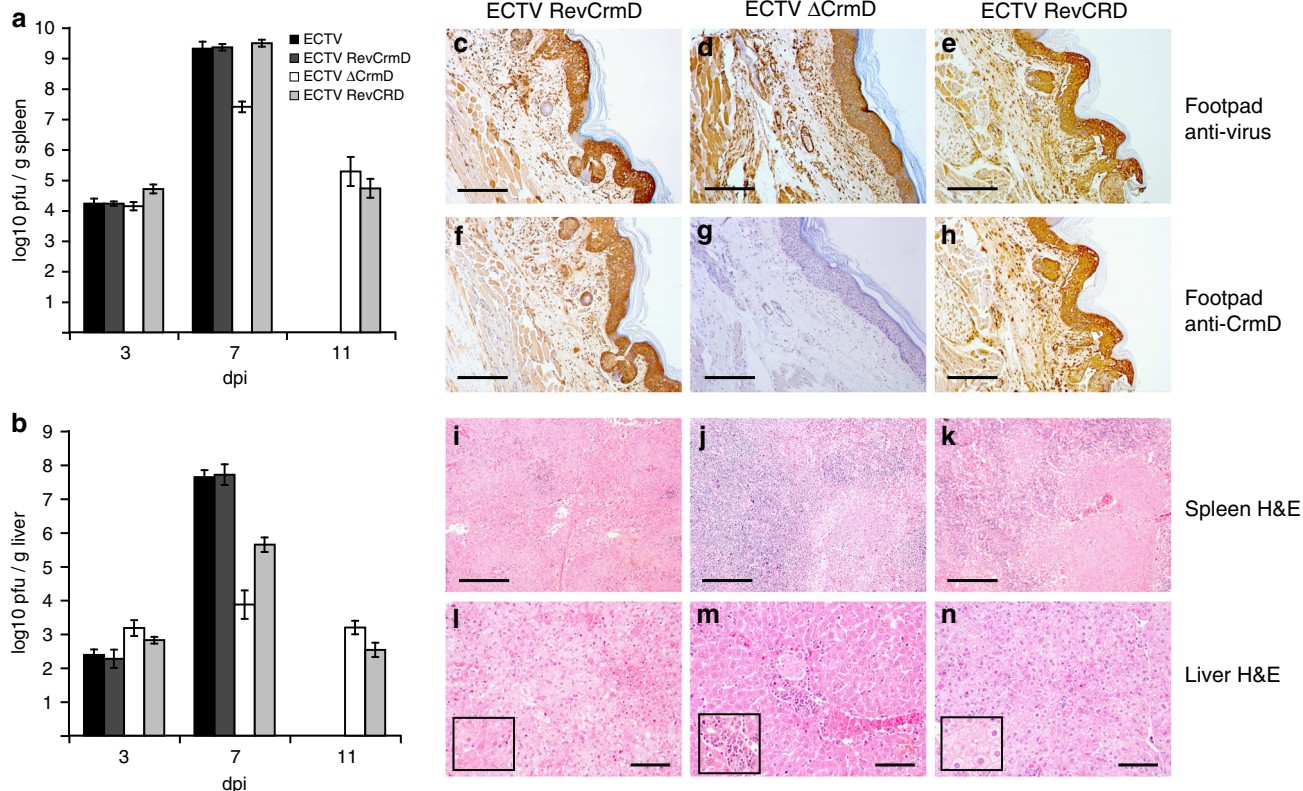

**Fig. 5** Impaired virus spread in the absence of ECTV CrmD. Graphs show viral titres at 3, 7 and 11 dpi in spleen (**a**) and liver (**b**) of BALB/c mice infected with 1000 PFU per animal of the indicated viruses. Data are mean log +/− SEM of groups of five animals for each condition. Note that mice infected with ECTV or ECTVRevCrmD did not survive up to 11 dpi. Left hind foot anti-poxvirus staining (**c−e**) or anti-CrmD staining (**f−h**) of zinc-fixed footpad sections of representative BALB/c mice infected with 1000 PFU of ECTVRevCrmD (**c, f**), ECTVΔCrmD (**d, g**) or ECTVRevCRD (**e, h**) at 7 dpi. H&E staining of spleen (**i−k**) or liver (**l−n**) sections of representative mice infected with 1000 PFU of ECTVRevCrmD (**i, l**), ECTVΔCrmD (**j, m**) or ECTVRevCRD (**k, n**) at 7 dpi. Insets show enlargement of selected areas to illustrate necrosis (**l, n**) and inflammatory infiltrate (**m**). Data are from one experiment representative of two independent experiments. Scale bar, 100 μm (**c−k**) and 40 μm (**l−n**)

the secondary organs are required to survive lethal mouse-pox[21,22]. To understand the mechanisms by which ECTV CrmD allows viral replication and whether it impairs the immune response, we studied the early NK cell response and the late T cell response against the different recombinant viruses in the DPLN at 2 dpi and in the spleen at 7 dpi, respectively.

The NK cell population (CD3−DX5+) of all the infected groups represented around 3% of the cells in the DPLN whereas it was less than 1% in the PBS-inoculated group (Fig. 8a, c). Furthermore, no differences in the total number of NK cells present in the DPLN at 2 dpi were found among the different infected groups (Fig. 8b). This suggested that CrmD is not involved in controling the recruitment of NK cells to the early virus replication sites. However, differences in the activation status of NK cell populations were found. In the PBS-inoculated group less than 2% of NK cells were activated, as assessed by granzyme B expression (Fig. 8d). Conversely, in mice infected with recombinant viruses lacking the TNF blocking ability, ECTVΔCrmD and ECTVRevSECRET, more than 20% of NK cells were activated. In contrast, viruses expressing the TNF binding domain, ECTVRevCRD and ECTVRevCrmD, significantly controlled NK cell activation, reducing by half the % of granzyme B+ NK cells (Fig. 8d). These results suggested that the anti-TNF activity of CrmD impairs the early NK cell activation in response to ECTV infection.

At 7 dpi, infection with the CrmD-expressing virus produced an almost complete elimination of CD8 T cells from the spleens (Fig. 8a, e). It is important to clarify that although the

representative dot blot in Fig. 8a (bottom panel) still shows a 1.9% of CD8 T cells in the spleen of ECTVRevCrmD-infected mice, this was only slightly above the staining observed with the corresponding isotype control (1.3%). In addition, only 40% of the analyzed cells fell inside the lymphocyte gate in this group, whereas the analysis gate gathered more than 80% of the cells in all the other groups (Supplementary Fig. 3). These two factors explain the almost complete depletion of CD8 cells in mice infected with ECTVRevCrmD shown in Fig. 8e. This splenic lymphopenia has been observed previously in ECTV lethal infections[41–43]. In mice infected with either ECTVΔCrmD, ECTVRevCRD or ECTVRevSECRET, however, CD8 T cells were detectable and efficiently activated in response to infection (Fig. 8a, e and f). This showed that the presence of both the TNF and chemokine binding domains in the CrmD protein is necessary for the inhibition and elimination of CD8 T cells in the spleen, and for full virulence of ECTV.

**Immunization with ECTV CrmD protects from fatal mouse-pox.** As ECTV infection was severely attenuated in the absence of ECTV CrmD, we hypothesized that a blockade of ECTV CrmD protein may prevent the development of mousepox. To test this, we immunized susceptible mice with purified recombinant ECTV CrmD protein and challenged them with a lethal dose (100-fold LD50) of ECTV, to test the induction of an efficient protective response. Sera from CrmD-immunized mice (14 out of 15 mice), but not from control mice, neutralized the ability of CrmD to

**Table 2 Histopathology of spleen and liver in recombinant ECTV-infected mice**

|  | 3 dpi | | 7 dpi | | 11 dpi | |
| --- | --- | --- | --- | --- | --- | --- |
|  | Spleen[a] | Liver[a] / [b] | Spleen[a] | Liver[a] / [b] | Spleen[a] | Liver[a] / [b] |
| EV | − | − / − | +++ | + / + | n.a. | n.a. |
| EV RevCrmD | − | − / − | +++ | + / + | n.a. | n.a. |
| EV ΔCrmD | - | − / − | + | − / ++ | + | − / ++ |
| EV RevCRD | - | − / − | ++ | + / + | + | − / ++ |

Groups of five BALB/c mice were infected s.c. in the left hind footpad with the indicated viruses and sacrificed at different dpi. Spleen and liver were H&E-stained for histopathological analysis. Degree of necrosis[a] was semiquantitatively assessed using an arbitrary scale: − negative findings (0%); + slight (about 25% necrosis); ++ moderate (about 50% necrosis); +++ very intense (about 90–100% necrosis). Presence of inflammatory infiltrate[b] was evaluated in a minimum of 10 fields per liver slice section to obtain the mean value and it was scored as: − negative findings; + slight; ++ moderate; +++ very intense. n.a. not applicable

inhibit TNF activity in a cytotoxicity assay (Figs. 9a, 2 µl dose) causing <50% cell viability. Addition of a lower amount of sera in the TNF biological assay identified a weaker CrmD neutralization activity in three of the five mice that succumbed to infection after CrmD immunization (Figs. 9a, 1 µl dose). After ECTV inoculation, mice previously immunized with CrmD developed mousepox signs and suffered weight loss to a similar degree as those injected with PBS (Fig. 9b, c). However, the immunized mice showed very early and acute footpad swelling in response to the infection (Fig. 9d), reminiscent of that observed in ECTVΔCrmD-infected mice and suggesting that ECTV CrmD activity produced by WT ECTV was neutralized in these animals. Moreover, 67% of the CrmD immunized mice survived infection and had fully recovered by 30 dpi, while all the PBS injected mice had died by 10 dpi (Fig. 9e).

## Discussion

The results presented here show that ECTV CrmD is an essential factor for mousepox virulence and that both its TNF and chemokine inhibitory activities contribute to its immunomodulatory role. ECTV replication in secondary replication organs, spleen and liver, was impaired in the absence of CrmD. The most probable cause of death during mousepox is liver failure due to extensive viral replication[32]. Therefore, reduced viral replication and hence necrosis in this organ may account for survival of mice infected with ECTV lacking either the full length CrmD, or the anti-TNF or anti-chemokine activities of the protein. Additionally, expression of CrmD in the spleen and liver will block anti-viral responses in these organs, consistent with an increased inflammatory response in the liver of ECTVΔCrmD- and ECTVRevCRD-infected mice.

Deletion of the *CrmD* gene from ECTV results in one of the most profound effects on virulence described in poxviruses[17,18]. Previous reports have shown that inactivation of other vTNFRs secreted by VACV, CPXV and myxoma virus causes limited viral attenuation, and some reports are not conclusive[35–37]. Similarly, inactivation of vCKBPs, 35-kDa and A41 proteins, from poxvirus genomes causes increased leukocyte recruitment to the sites of infection without major effects on the disease progression or mortality, or a slight attenuation as a result of reduced inflammatory pathology[17,18]. Interestingly, deletion of the ECTV type I IFN binding protein, which targets a different cytokine, rendered the virus avirulent to a degree similar to that we observed after CrmD deletion[44]. This suggests a possible link between the type I IFN and the TNF/LT/chemokine anti-viral host responses, as the link described for the type I IFN and nuclear factor kappa B pathways[45]. Increased LT signalling in the absence of CrmD could contribute to the induction of the IFN response, as described for murine cytomegalovirus[46], restricting virus replication through increased type I IFN signalling to immune cells or direct effects on infected cells. In accordance with this, TNF and

IFN may act synergistically in anti-viral defense[47]. Also, a role of the LT network in controlling the type I IFN response has been proposed[48].

Previous data suggested a role for TNF-induced signalling in mousepox pathogenesis, as transgenic resistant mice lacking functional TNFR1 and TNFR2 became susceptible to ECTV[29]. Consistently, treatment of susceptible BALB/c mice with murine TNF hindered ECTV replication and mortality to some extent[33]. However, TNF had a relatively minor role during ECTV or related poxvirus infections in terms of impact on LD50. Strikingly, the lack of the secreted TNF binding protein CrmD has a profound effect, reducing virulence almost completely. While specific experimental setups may account for these differences, they indicate a role of CrmD beyond inhibition of soluble TNF, which may include inhibition of LTα, LTβ or chemokines. In addition, we have recently demonstrated that CrmD and other vTNFRs interact with membrane TNF and inhibit its cytotoxic activity[14]. Further, CrmD may trigger reverse signaling in membrane TNF-bearing cells, as shown for viral CD30[7], and this may influence the anti-viral response.

This CrmD-membrane TNF interaction might also explain the impaired activation of NK cells in the DPLN of mice infected with ECTVRevCrmD and ECTVRevCRD. NK cells are required at early stages to curb ECTV dissemination from the lymph nodes to the spleen and liver and ultimatelly, to survive to fatal mousepox[41,49]. Despite historically considered effector cells of the innate immunity, NK cell responses are greatly modulated by dendritic cells (DC)[50,51], being more efficient when both cells are in direct contact[50,51], and this is important for defense against other viral infections[52]. Accordingly, DC-depleted C57BL/6 mice are susceptible to ECTV[53]. The DC-NK crosstalk is mediated in mouse and human by the engagement of DC membrane TNF, but not soluble TNF, with NK TNFR2[54,55]. Therefore, CrmD or the truncated TNF binding domain may be blocking the membrane TNF-TNFR2 interaction hindering an efficient DC-NK cell crosstalk. A weak NK cell activity might explain the high viral titers of ECTV and ECTVRevCRD detected later in the spleen, whereas the dissemination of ECTVRevSECRET and the CrmD deletion mutant is curbed by a potent early NK cell response. After 4 dpi NK cells are no longer required for resistance to ECTV challenge and the T cell response takes over[41]. At day 7 in the spleen, we observed a depletion of CD8 lymphocytes in mice infected with ECTV expressing CrmD, however, an efficient CD8 response was mounted against all the other viruses tested here. Of note, this CD8 cell depletion did not correlate with viral loads in the spleen since ECTVRevCRD, which reaches equally high viral titers, did not cause lymphopenia. This result suggests that both CrmD immunomodulatory activities, anti-TNF and anti-chemokines, are required for complete control of the T cell response in the spleen. Consistently, the SECRET domain binds Cxcl11, one of the ligands of the chemokine receptor CXCR3 that enhances the ability of CD8 T cells to locate VACV-infected cells

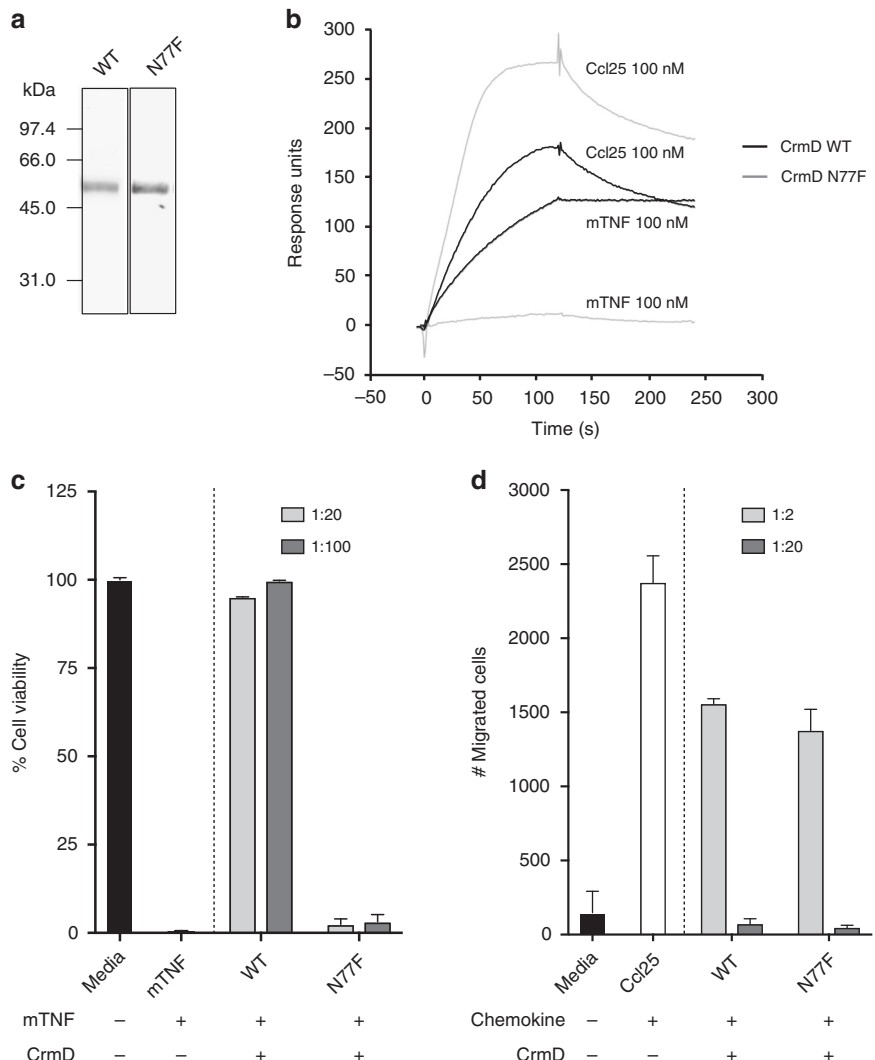

**Fig. 6** CrmD N77F mutant blocks chemokines but lacks anti-TNF activity. **a** Coomasie blue stained gel showing 500 ng of recombinant CrmD wild type (WT) and N77F mutant expressed by recombinant baculoviruses. Molecular mass standards are shown in kDa. **b** CrmD N77F binds mouse Ccl25 but not mouse TNF (mTNF). The binding of 100 nM Ccl25 and mTNF to CrmD WT or N77F mutant was assessed by SPR experiments. **c** CrmD N77F mutant does not interfere with mouse TNF-mediated cytotoxicity on L929 cells. The cell viability of L929 cells after a 18 h incubation with 1.2 nM TNF in the absence or presence of the indicated molar ratios of CrmD variants was determined. Data is represented as mean ± standard deviation (SD) of the % relative to cells incubated without TNF (media). **d** CrmD N77F blocks mouse Ccl25-induced migration of MOLT-4 cells. The number of cells that migrated through the transwell filter after 4 h incubation at 37 ºC with 50 nM mouse Ccl25 in the absence or presence of the indicated increasing molar ratios of CrmD variants is shown. Data is represented as mean ± SD of triplicates. In **c** and **d**, one experiment representative of three independent experiments is shown

and to exert anti-viral effector functions[56]. In agreement with a role of both TNF and chemokines in T cell responses, neutralization with anti-TNF antibodies does not affect the splenic late CTL response against ECTV in C57BL/6 mice[32]. However, this might be an indirect effect since splenic lymphopenia has also been observed in other ECTV lethal infections where the host ability to mount an efficient TNF and chemokine response is intact[41,42].

Here we show that ECTV CrmD efficiently inhibits the establishment of a proinflammatory state at the site of virus inoculation, as evidenced by the increased ICAM-1 expression observed in its absence. The ability of CrmD to block TNF and LT activity, cytokines that induce the expression of ICAM-1 on endothelial cells, could account for this[15,16]. In the absence of CrmD, an increased inflammatory infiltrate was observed both in the footpad and in the liver, consistent with the chemokine inhibitory function of its SECRET domain. More specifically, an

important fraction of the footpad infiltrate was composed of cells bearing the chemokine receptor CCR10, which supports migration of lymphocytes towards Ccl28 and Ccl27. Both chemokines are bound with high affinity by CrmD and expressed by skin keratinocytes in response to TNF and other proinflammatory stimuli[16,57]. Indeed, Ccl27 is important in T-cell mediated skin inflammation in vivo[58]. Thus, the potent anti-inflammatory activity of CrmD may be due to its ability to inhibit either TNF/LT as well as a set of chemokines. Consistent with this, VARV CrmB, with properties similar to CrmD, blocks cell migration induced after epicutaneous application of murine TNF[59].

The infection of mice with ECTVRevCRD or ECTVRevSE-CRET expressing the anti-TNF/LT or anti-chemokine activity of CrmD, respectively, allowed us to address the contribution of TNF/LT vs. chemokines to inflammatory and protective responses in vivo. Interestingly, ECTVRevCRD was not able to efficiently control the inflammatory infiltrate observed in

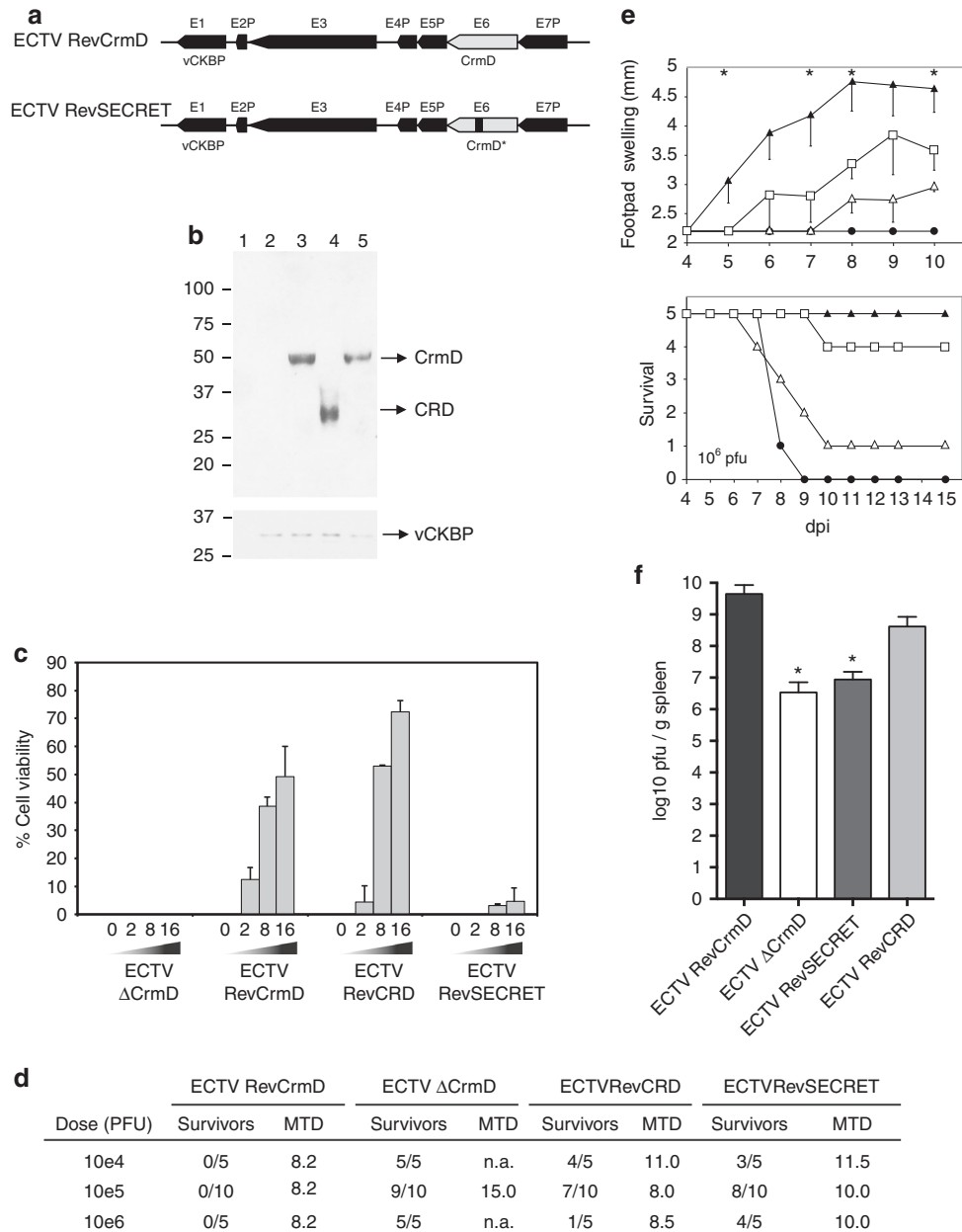

**Fig. 7** The activity of the SECRET domain in vivo is dependent on concomitant TNF blockade. **a** Schematic diagram of the genomic structure of ECTV RevSECRET. The genes flanking the *CrmD* locus are indicated. In gray, the *CrmD* locus in the parental and recombinant virus is shown. Presence of a point mutation (N77F) in the CrmD of ECTVRevSECRET is indicated by an asterisk and a black line. Only the left inverted terminal repeat is shown. **b** Western blot analyses using anti-CrmD and anti-35-kDa vCKBP antisera of supernatants from BSC-1 cells mock-infected (1) or infected with ECTVΔCrmD (2), ECTVRevCrmD (3), ECTVRevCRD (4) or ECTVRevSECRET (5) at a multiplicity of infection of 5 PFU/cell and harvested at 24 h post-infection. The position of the proteins is indicated by arrows. Molecular size markers in kDa are shown on the left. **c** TNF-induced cytotoxicity assay. Increasing amounts of supernatants (equivalent to 2, 8 or $16 \times 10^3$ cells) from cells infected with the indicated viruses were added to block the effect of TNF on L929 cells. Values of % cell viability obtained in the presence of mock-infected cell supernatants were subtracted from the % cell viability caused by the corresponding supernatant volume from virus-infected cells. Data are mean +/− SD of triplicate samples. **d** Mortality rate determination of CrmD recombinant ECTVs in susceptible mice. Groups of five or ten female BALB/c mice were infected s.c. in the left hind footpad with different doses of the indicated viruses. The number of survivors at 15 dpi and the mean time to death (MTD) in days for each condition are shown. n.a., not applicable. **e** Groups of five female BALB/c mice were inoculated s.c. in the left hind footpad with $10^6$ PFU of ECTV RevCrmD (black circles), ECTVΔCrmD (black triangles), ECTVRevCRD (white triangles) or ECTVRevSECRET (white squares) and monitored daily for mortality and footpad swelling, as indicated. Data are shown as mean +/− SD. Asterisks indicate time points at which significant differences ($p < 0.01$, multiple *t* tests with false discovery rate correction at $Q = 1\%$) between ECTVΔCrmD and ECTVRevSECRET-inoculated mice were found. **f** Replication of recombinant ECTVs in spleen. Viral titres at 7 dpi in spleen of mice infected with $10^5$ PFU per animal of the indicated viruses. Data are mean log +/− SD of groups of five mice for each condition. Groups significantly different from the ECTVRevCrmD-infected group are indicated (asterisks $p < 0.05$, ANOVA with Bonferroni multiple comparison test)

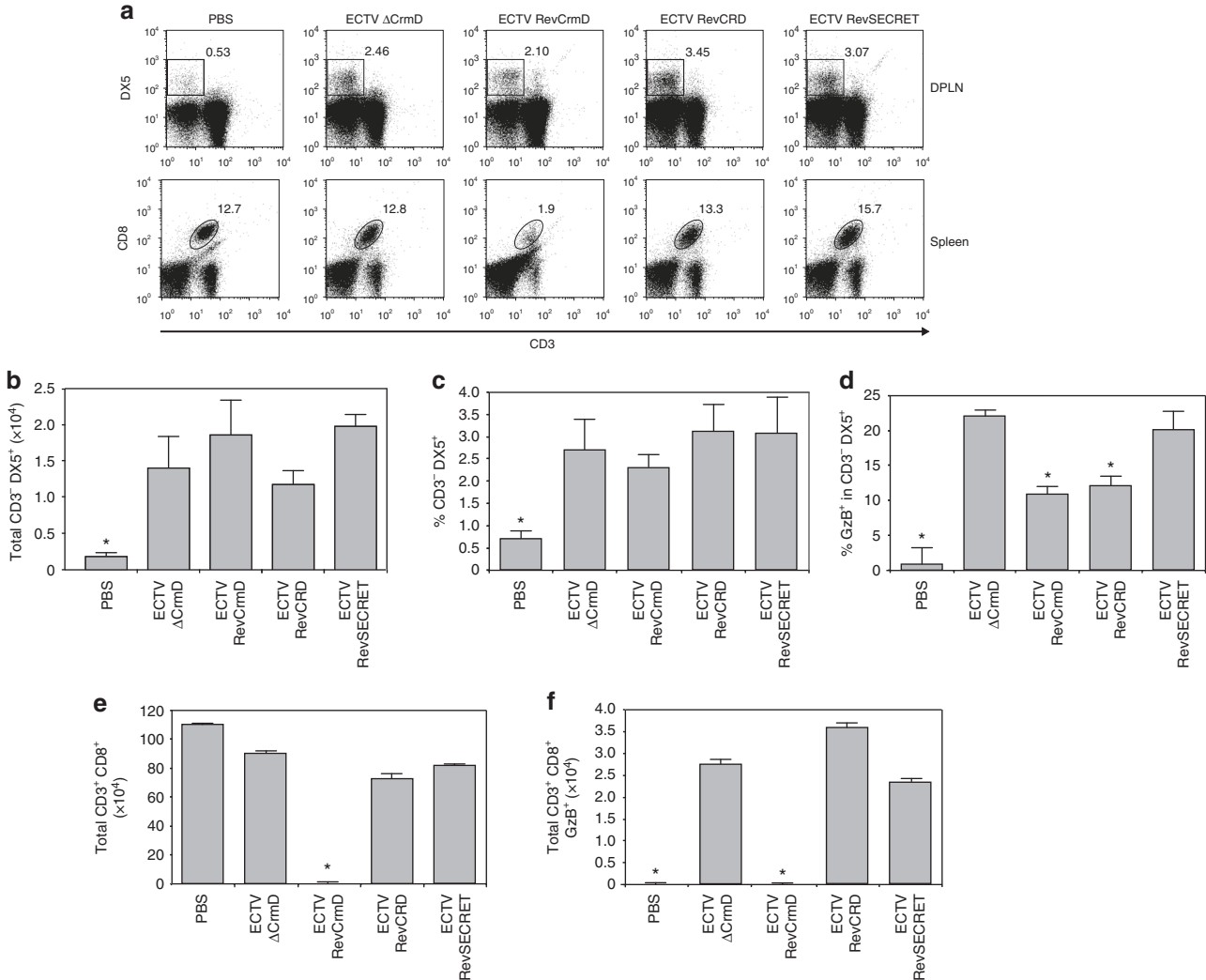

**Fig. 8** Inhibition of TNF activity in vivo impairs NK cell activation in response to ECTV infection. Cells from DPLN (**a–d**) harvested at 2 dpi or from spleens (**a, e, f**) collected at 7 dpi from PBS-inoculated BALB/c mice or mice infected with $10^5$ pfu of the indicated viruses were analyzed by flow cytometry using conjugated anti-CD3, anti-CD8, anti-DX5 and anti-granzyme B (GzB) antibodies. In **a**, representative dot plots of the staining of NK cells (CD3$^-$ DX5$^+$; top panel) and CD8 T cells (CD3$^+$CD8$^+$; bottom panel) isolated from DPLN or spleen, respectively, are shown. Number inside each graph indicates the % of cells inside the depicted gates. In **b** and **c**, a quantification of the total number and % of NK cells, respectively, is presented. The % of NK cells expressing granzyme B in each group is quantified in **d**. In **e** and **f**, a quantification of the total number of CD8 T cells and granzyme B-expressing CD8 T cells detected for each group is shown. The number of events positively stained with the corresponding isotype control antibodies (isotype DX5, 0.15% positives; isotype CD8, 1.3% positives; isotype granzyme B, 0.02% and 0.03% positives in DPLN and spleen, respectively) were subtracted from each sample for the quantification analyses. Data are mean $+/-$ SD from one experiment representative of three independent experiments with 4–5 animals per group. Statistically significant groups are indicated (asterisks $p < 0.05$, ANOVA with Bonferroni multiple comparison test)

ECTVΔCrmD-infected mice, in spite of the well-documented and potent pro-inflammatory function of TNF. This suggested that the activity of the mouse chemokines targeted by the SECRET domain (Ccl24, Ccl25, Ccl27, Cxcl11, Cxcl12β, Cxcl13 and Cxcl14) was sufficient to control cell migration and to trigger inflammation when TNF was neutralized by the truncated CrmD protein. The delay in the appearance of the inflammatory infiltrate in ECTVRevCRD-infected mice as compared to that observed with ECTVΔCrmD, probably reflects TNF blockade at the inoculation site and suggests the involvement of TNF in triggering the initial response in vivo. It is important to note that the infiltrating cells are probably contributing to an increased expression of cytokines, and thus chemokine blockade by the CrmD SECRET domain expressed in ECTVRevSECRET infections may impair localized TNF expression. Limited recruitment of cytokine producing cells into the sites of infection may be a general principle in host-pathogen interactions and help pathogens escape the host response, as shown in a *Leishmamia major* mouse model of dermal infection[60] and in *Listeria monocytogenes*-infected mice, where Ccl2-induced recruitment of TNF and iNOS producing cells to the spleen mediates an effective innate immune response[61].

The reasons for the dramatic attenuation phenotype of the CrmD mutant could be related to the nature of the chemokines targeted and/or the inhibitory mechanism. The CrmD SECRET domain is different from the 35-kDa vCKBP in the specific set of chemokines blocked[17,18]. The VACV A41 protein and its ECTV Naval orthologue E163 bind a number of chemokines including those recognized by the SECRET domain, but they do not block chemokine interaction with its receptor and are rather proposed to dissipate chemokine chemotactic gradients in vivo by targeting the glycosaminoglycan binding site of chemokines[17,18]. Thus the

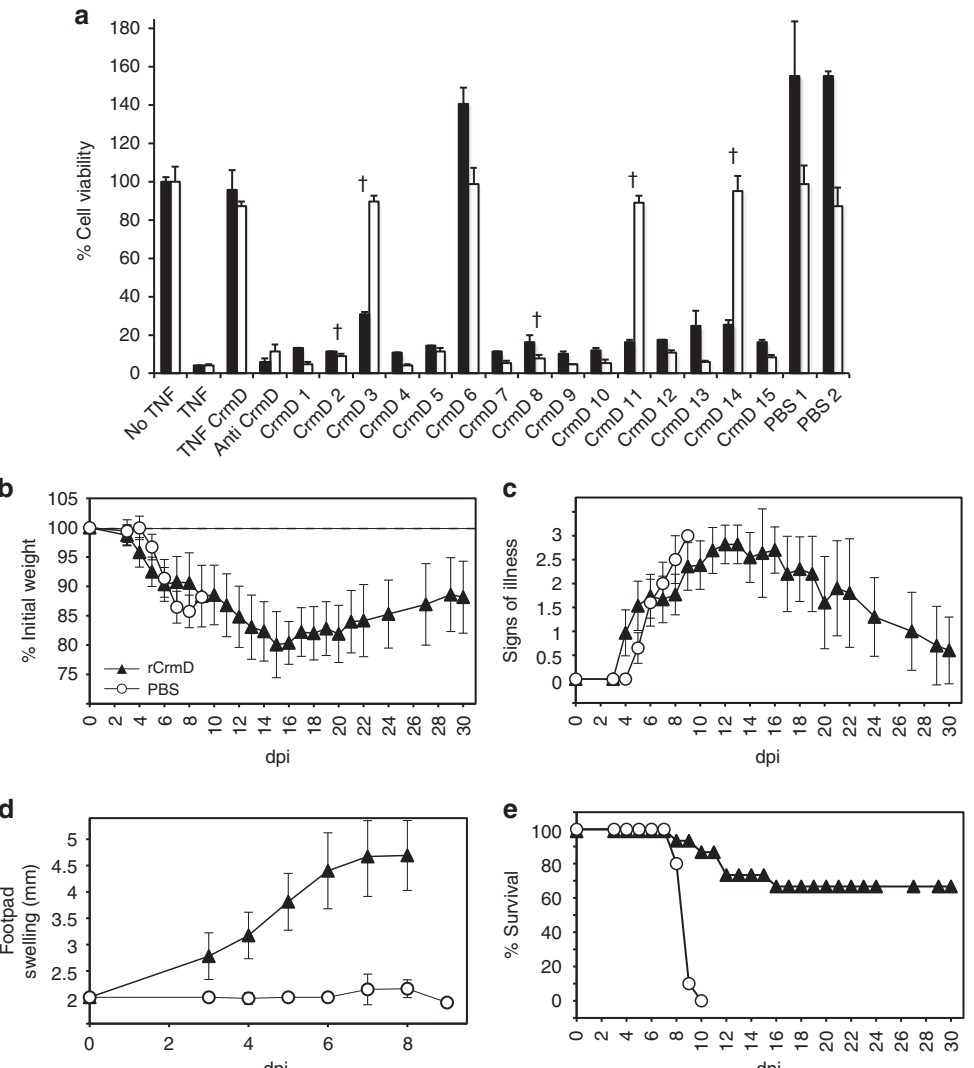

**Fig. 9** Immunization with purified recombinant ECTV CrmD protects mice from fatal mousepox. Groups of BALB/c mice were immunized with PBS ($n = 10$) (PBS, white circles) or recombinant ECTVCrmD protein ($n = 15$) (rCrmD, black triangles), and challenged with 1000 PFU of ECTV per animal at 18 days after concluding the immunization procedure. **a** Neutralization of TNF activity in a TNF-induced cytotoxicity assay by serum from mice immunized with CrmD or PBS. L929 cells were incubated with TNF and CrmD in the presence of 1 µl (white bars) or 2 µl (black bars) of serum from mice immunized with CrmD (CrmD 1–15) or PBS (PBS 1–2). As controls, cell viability was determined in the absence of TNF (no TNF) or in the presence of TNF (TNF) incubated also with CrmD (TNF CrmD) or an antiserum against CrmD (anti CrmD). Mice that succumbed to infection are indicated with a cross. Data are mean +/− SEM of triplicate samples. **b**–**e** Mice infected with ECTV were monitored for weight loss (**b**), signs of illness (**c**), footpad swelling (**d**) and mortality (**e**) at the indicated times of infecion. Data are shown as mean +/− SEM

35-kDa vCKBP and A41 ECTV orthologues are probably involved in controlling different aspects of the anti-viral host response. Moreover, ECTV encodes two SECRET domain-containing proteins, named E12 and E184, that show the same chemokine binding specificity[16] but these proteins are not able to compensate for the loss of the CrmD SECRET domain. The phenotype of ECTVrevSECRET, expressing anti-chemokine activity but lacking TNF/LT inhibitory activity, indicates that the SECRET domain needs simultaneous TNF blockade to have a major impact on virus virulence. Additionally, the finding of the SECRET domain on the same molecule than a TNF binding domain might be relevant to ensure an efficient combined effect in the infected host. It is reasonable to propose that the coordinated blockade of cytokines and chemokines by a viral protein could serve as an excellent inhibitor of cell recruitment and the immune response in vivo, as shown here for the poxviral CrmD protein and previously for murine cytomegalovirus[62].

The study of the contribution of CrmD to mousepox sheds light into the mechanisms of the pathogenesis of acute viral infections. Importantly, mousepox is a model for human small-pox, a severe human disease caused by VARV and whose pathogenesis has not been extensively studied at the molecular level. VARV and ECTV share secreted immunomodulatory proteins and encode only one vTNFR[13,16]. The VARV protein CrmB has TNF/LT and chemokine inhibitory properties similar to those of ECTV CrmD, but is better adapted to block the human immune system[15,16]. The control of TNF-induced gene expression observed in VARV-infected monkeys suggests that CrmB is expressed during infection[63]. Our results suggest that VARV CrmB is an important virulence factor in smallpox. The same may be true for MPXV, an orthopoxvirus causing a smallpox-like disease in humans with mortality rates of up to 10% that encodes a CrmB orthologue[64]. The fear of either an intentional release of VARV or the emergence of a poxviral zoonosis,

caused by MPXV or other poxviruses[25,26] in a large unvaccinated population have sparked the interest in developing safer vaccines or drugs to prevent or treat human poxviral infections. Although current vaccines have a proven efficacy record, potential risks relating to secondary effects, the increased number of immuno-compromised individuals and uncertainty about the duration of protective immunity exist. Thus, alternative strategies for the development of safer vaccines are being currently pursued (reviewed in[65]). Recombinant protein is thought to be safer, and different combinations or single protein vaccination using components from virus particles, which induce neutralizing antibodies, protect from poxvirus infections[66–68]. Our report shows that immunization with recombinant ECTV CrmD protects mice from a lethal mousepox challenge, possibly by an antibody-mediated blockade of CrmD-ligand interaction and clearance of the CrmD protein from the infected host. Similar results have been obtained by vaccinating with the type I IFN binding protein from ECTV[44], and antibodies against this secreted viral protein protect from mousepox[69]. Thus, secreted poxviral immunomodulators can act as effective subunit vaccines in preventing a lethal poxvirus infection in its natural host and could be used singly or in combination with other proteins. It is important to note that the smallpox VACV vaccines Dryvax and Modified VACV Ankara do not express the CrmB protein and will not induce a neutralizing response against CrmB expressed by VARV or MPXV[13]. However, the Dryvax vaccine induced an immune response against viral structural proteins that was sufficient to protect from smallpox.

TNF and chemokines are important in the development of human pathologies unrelated to viral infection. Notably, soluble TNFRs are used to treat a variety of inflammatory conditions in the clinic such as rheumathoid arthritis, ankylosing spondylitis or psoriatic arthritis[70]. To date soluble versions of the human TNFR2 or monoclonal anti-TNF antibodies are used, although other strategies are under development. The use of vTNFRs in this context has been proposed[9,71], and transgenic mice expressing ECTV CrmD have shown that it can inhibit TNF driven inflammatory reactions in vivo[72]. The fact that CrmD acts as a potent anti-inflammatory molecule in vivo and that its SECRET domain is important for this activity suggests that addition of a chemokine inhibitory domain to the human soluble TNFRs may increase their clinical efficacy in certain settings. However, this approach should be taken with caution because an antirheumatic drug combining anti-TNF and anti-chemokine activities could be expected to further dampen the already debilitated immune response of patients under anti-TNF therapy, what might worsen the frequent infectious complications observed in these patients[73].

In conclusion, the characterization of the role of CrmD in mousepox pathogenesis has demonstrated a critical role of TNF and a specific set of chemokines in defense from virulent poxvirus infections. The nature of the ligands of CrmD may point to the cytokines and chemokines important in the control of poxviral infections.

## Methods

**Cells and viruses.** A plaque-purified and fully sequenced ECTV Naval isolate (Naval.Cam) was grown in BSC-1 cells (ATCC CCL-26)[24]. For titration of virus in organs from infected mice, spleen and liver were aseptically removed, weighed and homogeneized and serial dilutions plated on BSC-1 cell monolayers. For infection of mice, virus stocks were semipurified by centrifugation through a 36% sucrose cushion[74]. Viral stocks were routinely tested for the absence of mycoplasma and the endotoxin levels detected usingToxiSensor Chromogenic LAL Endotoxin Assay kit (GenScript) were under 0.3 EU/ml.

**Expression and purification of recombinant CrmD proteins.** For the generation of anti-CrmD polyclonal rabbit antibodies and immunization experiments, rabbits and mice, respectively, were injected with a recombinant CrmD fused to the Fc portion of a human IgG1 (CrmD-Fc). CrmD-Fc was expressed by recombinant baculoviruses and purified from the supernant of infected Hi5 insect cells (ThermoFisher BTI-TN-5B1-4) by affinity chromatography in a protein A coupled sepharose column. Similarly, a CrmD C-terminally tagged with V5 and 6xHis epitopes was expressed by recombinant baculoviruses as previously described[15]. A N77F mutant of this CrmD-V5-6xHis protein was generated using the Quik-Change II Site-Directed mutagenesis kit (Agilent Technologies) and expressed by recombinant baculoviruses[15].

**CrmD anti-TNF activity assay.** The ability of ECTV CrmD to block TNF-induced cell death was determined as described[16]. Briefly, L929 cells (ATCC CCL-1) were incubated with TNF (R&D Systems, Minneapolis, USA) which had been pre-incubated or not with recombinant ECTV CrmD protein or supernatants from ECTV-infected cells and cell death determined using the CellTiter OneSolution viability assay (Promega).

**Chemotaxis assays.** The anti-chemokine activity of CrmD wild type and the N77F mutant was assessed by chemotaxis assays using a 96-well ChemoTx plate with a 3-μm pore sized filter as previously described[16]. Briefly, MOLT-4 cells (ATCC CRL-1582) were incubated with 70 nM of mouse Ccl25 (Peprotech Inc., London, UK) in RPMI 0.1% FBS in the presence or absence of CrmD wild type or N77F at the indicated molar ratios. Unspecific migration in the absence of chemokine was also monitored as reference (media). Cell migration through the filter was allowed to occur during 4 h at 37 °C, and subsequently the number of cells that migrated to the bottom well was calculated by interpolation in a standard curve of number of cells using CellTiter Aqueous One Solution assay kit (Promega).

**Surface plasmon resonance.** The binding affinity of recombinant CrmD for mouse chemokines (Peprotech Inc., London, UK) and the ability of wild type CrmD and mutant N77F to interact with mouse TNF (R&D Systems, Minneapolis, USA) and Ccl25 (Peprotech Inc., London, UK) were determined by SPR using a Biacore X biosensor (GE Healthcare).

For affinity determinations, recombinant CrmD was immobilized onto a flow cell of a CM4 chip at low density (900 RUs) by the amine coupling protocol. One flow cell was left empty to be used as reference. Increasing concentrations of mouse chemokines were injected in HBS-EP buffer (GE Healthcare) over the chip surface and their binding was recorded for 120 s followed by a 300 s dissociation period. Surface was regenerated with glycin-HCl pH2.0 between injections. Binding sensorgrams were analyzed by the BiaEvaluation software (GE Healthcare) and fitted to a general 1:1 Langmuir binding model.

For binding assays, recombinant wild type and N77F CrmD proteins were immobilized on to a flow cell of a CM4 chip at high density (1500 RUs) as explained above. 100 nM of mouse TNF or mouse Ccl25 were injected over the chip in HBS-EP buffer and their association was monitored during 120 s followed by a 120 s dissociation. Binding sensorgrams were processed and analyzed using the BiaEvaluation software.

**Construction of recombinant ECTVs.** Recombinant ECTVs were generated using a transient dominant selection procedure and the selection in the presence of puromycin as previously described[74]. The plasmid pMS30 was constructed for expression of EGFP under a VACV early-late promoter followed by an IRES cassette for expression of the puromycin acetyl transferase gene from the same transcript. For the generation of ECTVΔCrmD, the flanking regions of the CrmD gene were PCR-amplified and cloned into the EcoRI and PstI restriction sites of the polylinker region of pMS30. The 5′ flanking region of the CrmD gene was amplified with oligonucleotides CrmD-27 (5′-GCGGAATTCCGATTTAATAACATTC GATTATATAG) and CrmD-11 (5′-CGCGGATCCGGTGTATACGGAA-CATCTCCAC), and the 3′ flanking region of CrmD was amplified with oligonucleotides CrmD18 (5′-CGCGGATCCTAACATGGACGTCGTCGCGTATCATAC) and CrmD28 (5′-GCGCTGCAGCTCTGTAATGATGGACGTTATTTC), to generate the plasmid pMS34 (pΔCrmD). Both flanking regions and the CrmD gene were PCR-amplified with oligonucleotides CrmD27 and CrmD28 to generate the plasmid pMS37 (pRevCrmD) that was used for reinsertion of the CrmD gene into the ECTVΔCrmD genome and construction of ECTVRevCrmD. The 5′flanking region used for generation of ΔcrmD and the TNF binding domain of CrmD including a stop codon were PCR-amplified using oligonucleotides CrmD27 and CrmD30 (5′-CGCGGATCCTAACAAGAGGTCTTGTTAACAGGATAC) and pMS37 as a template. The resulting PCR product was cloned into the EcoRI and BamHI sites of pMS34 generating plasmid pAH7. This plasmid was used to generate ECTVRevCRD, which expresses a truncated version of CrmD corresponding to residues M1 to C180. For the generation of ECTVRevSECRET, the CrmD gene contained in pMS37 was mutated by directed single point mutation using the QuickChange II mutagenesis kit (Agilent Technologies) and the primers CrmD43 (AGATGACACCTTTACATCCATTCCTTTTCATAGTCCCGCGTG) and CrmD44 (CACGCGGGACTATGAAAAGGAATGGATGTAAAGGTGTCATCT). These primers introduce a N77F mutation in CrmD.

After transfection/infection in BSC-1 cells, the intermediate single-crossover recombinant viruses in which the complete plasmid has been inserted into the ECTV genome were selected for three to five consecutive infection rounds in the

presence of puromycin and monitored for EGFP expression by fluorescence microscopy. Recombinant viruses (ECTVΔCrmD, ECTVRevCrmD, ECTVRevCRD and ECTVRevSECRET) were finally selected by successive plaque purification of white plaques in the absence of puromycin and screening with a *CrmD*-specific PCR. The complete genome sequence of ECTVRevCRD and ECTVRevSECRET was determined by Illumina sequencing to confirm the genomic structure and the absence of inadvertent mutations that may affect virus virulence[24]. The sequences have been submitted to the European Nucleotide Archive and have been assigned reference number PRJEB19928. The number of sequencing reads that aligned with the ECTV Naval genome was $1.29 \times 10^6$ (93% of total sequencing reads) for ECTVRevCRD and $1.27 \times 10^6$ (87.4% of total sequencing reads) for ECTVRevSECRET. ECTVRevCRD was sequenced with a 325× coverage and, including the expected introduction of a truncated version of the duplicated *CrmD* gene, three changes were identified: Δ5.655-5.998 (*EVN006/CrmD* gene); T199.552 C (*EVN200P* pseudogene) and Δ201.621-201.964 (*EVN201/CrmD* gene). ECTVRevSECRET was sequenced with a 320× coverage and, including the expected N77F mutation in the amino acid sequence of duplicated *CrmD* gene, three changes were identified: TT6.310-6.311AA (*EVN006/CrmD* gene); T199.552 C (*EVN200P* pseudogene) and AA201.310-201.311TT (*EVN201/CrmD* gene). Both recombinant viruses had the expected mutations in both copies of the *CrmD* gene, present at the left and right ends of the viral genome. We also identified an additional point mutation in the inactive pseudogene EVN200P that was present in both ECTVRevCRD and ECTVRevSECRET, suggesting that this mutation was introduced during the generation of ECTVΔCrmD.

**Infection of mice.** Female BALB/c OlaHsd mice (6–8 weeks old) (Harlan), housed in ventilated racks, were anesthesized with isofluorane and s.c. infected in the footpad with 10 μl of virus inoculum. Viral doses were confirmed by titrating again on the same day the virus dilutions were used for mouse infections. Mice were housed in ventilated racks (Tecniplast) under biological safety level 3 containment facilities. Monitoring of infected animals was performed daily. Animals were weighed, scored for clinical signs of illness (scores ranging from 0 for healthy animals to 4 for severely diseased animals) and footpad swelling measured. Data analysis was performed using GraphPad Prism 6 (GraphPad Software, La Jolla, CA, USA). Survival curves were compared using the Logrank (Mantel-Cox) test. Footpad swelling and % initial weight data were analyzed using multiple t tests with false discovery rate $Q = 1\%$. Analyses were performed up to times post-infection at which survival rates in the corresponding groups were above 50%. ANOVA analyses with Bonferroni multiple comparison tests were performed in some experiments, as indicated, for comparisons among groups and times post-infection at which no mortalities were observed. These experiments were approved by the Biological Safety Committee of the Centro de Investigación en Sanidad Animal (CISA, INIA, Valdeolmos, Madrid) and animals were housed and handled according to legal requirements.

**Immunohistochemistry and semiquantitative analyses.** Footpad, spleen and liver samples from infected mice were removed aseptically and fixed in 10% buffered formalin solution to detect virus, ECTV CrmD protein and chemokine receptors, and in zinc fixative (BDPharmingen) to detect lymphoid cells and ICAM-1. After fixation, the samples were dehydrated through a graded series of alcohol to xylol incubations and embedded in paraffin wax. For structural and immunohistochemical analysis, sections (3 μm) were cut and stained with H&E or processed for immunohistochemical techniques. To detect virus and ECTV CrmD protein, formalin fixed serial sections were incubated with polyclonal rabbit anti-VACV antibody from a VACV-infected rabbit or a polyclonal rabbit anti-CrmD antibody against purified CrmD expressed in the baculovirus system. Both antibodies were generated in our laboratory. Secondary goat anti-rabbit IgG (Dako) was detected with an avidin-peroxidase-complex kit (PIERCE, Thermo Scientific) and 3,3′-diaminobenzidine tetrahydrochloride (Sigma) following the manufacturer´s instructions. The slides where counterstained with Mayer´s haematoxylin, dehydrated, and mounted with DPX mountant (Surgipath). Specific primary antibodies were replaced by PBS or normal goat serum in negative control sections. To detect lymphoid cells and chemokine receptors, the avidin-biotin alkaline-phosphatase staining method was used. Sections were dewaxed and immunostained with polyclonal rabbit anti-human CD3 (Dako), rat anti-mouse CD45R/ B220, CD4, CD8a, and CD8b (BD Pharmingen) or goat anti-mouse CCR10 (AbCAM). Anti-mouse ICAM-1 antibody was from AbCAM. For CD3 and CCR10 immunohistochemistry, antigen retrieval was achieved by heating sections in 0.1 M citrate buffer at pH 6. Secondary goat anti-rabbit IgG, rabbit anti-rat IgG, or rabbit anti-goat IgG (Dako) were used as corresponded with the streptavidin-biotin-alkaline phosphatase kit (PIERCE, Thermo Scientific) and "FastRed" (Fast red substrate packs, Lab Biogenex®) for detection of the immunogens, following the manufacturer's indications. The slides were counterstained with Mayer's haematoxylin, and mounted with Immu-mount (Thermo Shandon). Specific primary antibodies were replaced by PBS, normal rabbit serum or normal goat serum in negative control sections.

For semi-quantitative analyses of histological sections, samples from at least 5 animals for each parameter were analyzed in every case. To establish the degree of necrosis, a minimum of 10 fields were scored per spleen and liver slice to obtain the mean value. To quantify the morphological changes, sections were graded for necrosis using an arbitrary scale: – negative findings (0%); + slight (about 25% necrosis); ++ moderate (about 50% necrosis); +++ very intense (about 90–100 % necrosis). Inflammatory infiltration was evaluated in a minimum of 10 fields per liver slice or the complete footpad section to obtain the mean value and it was scored as: – negative findings; + slight; ++ moderate; +++ very intense. For antibody staining of lymphoid cells and CCR10 chemokine receptor expressing cells, all the cells from the inflammatory infiltrate were counted for each case and mean values are presented. In the case of anti-ICAM-1 staining, all the blood vessels from sections from ECTV- ($n = 5$), ECTVRevCrmD- ($n = 5$), and ECTVΔCrmD- ($n = 6$) infected mice were analyzed and scored as staining or non-staining. Mean percentage of staining vessels and standard deviations were calculated using the Excel spreadsheet and statistical significance was confirmed using a Student's t-test ($p < 0.01$).

**Flow cytometry.** DPLN and spleens from PBS-inoculated or ECTV-infected BALB/c mice were collected at 2 and 7 dpi, respectively, in RPMI supplemented with 10% FCS. Cell suspensions were obtained by homogenization of the organs through 40 μm cell strainers (BD Bioscience). Red blood cells were lysed by hypoosmotic shock in milli-Q water and white cells were washed twice in PBS and counted manually in a haemocytometer. DPLN cells were stained with anti-DX5-FITC (eBioscience), anti-CD3e-PerCP (eBioscience) and anti-GzB-APC (R&D Biosystems). Splenocytes were stained with anti-CD3e-PerCP, anti-CD8-PE (eBioscience) and anti-GzB-APC. In parallel, cell suspensions were also stained with the appropriate isotype control antibodies. 100,000 cells were analyzed in a FACS Calibur flow cytometer (Becton Dickinson). Events were gated according to a forward and side scatter pattern compatible with healthy lymphocytes (Supplementary Fig. 3). Results were analyzed with FlowJo software (FlowJo LLC).

**Immunization with recombinant purified ECTV CrmD.** A group of 15 female (6–8 week old) BALB/c mice was inoculated i.p. with 10 μg of purified recombinant ECTV CrmD protein expressed in the baculovirus system per animal three times at 17 to 20 day intervals. At 18 days after the last inoculation, mice were bled and sera obtained to check for presence of anti-CrmD antibodies and challenged s.c. with 1000 PFU of ECTV as above. As a control, a group of 10 BALB/c mice was subjected to the same protocol using PBS for i.p. inoculations and infected with equal amounts of virus. Disease progression was monitored as described above.

**Data availability.** The data that support the findings of this study are available from the corresponding author upon reasonable request. The viral genomic sequences reported have been submitted to the European Nucleotide Archive and have been assigned reference number PRJEB19928.

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

## Acknowledgements

We thank Javier Salguero for help with animal experimentation and immunohistochemistry, Rocío Martín and Carolina Sánchez for technical assistance and Daniel Rubio for discussions on the project. This work was funded by Grants from the Spanish Ministry of Economy and Competitiviness and European Union (European Regional Development's Funds, FEDER) (grant SAF2015-67485-R), and the Wellcome Trust (grant 051087/Z97/Z). M.B.R.-A. and A. Alejo were recipients of a Ramón y Cajal Contract from the Spanish Ministry of Science and Innovation.

## Author contributions

A. Alejo, M.B.R.-A. and A. Alcami conceived and designed the research; A. Alejo, M.B. R.-A. and S.M.P. performed most of the experiments; M.S. contributed to the construction of recombinant viruses; B.H. performed the genome sequence analysis of recombinant viruses; M.M.F.M. provided support for animal experiments and carried out histology and immunohistochemistry analyses. A. Alejo, M.B.R.-A., S.M.P. and A. Alcami wrote the manuscript. All authors discussed the results and commented on the manuscript.

## Additional information

**Competing interests:** The authors declare no competing interests.

