## [Peer Review File · Nature Communications]

Reviewers' comments:

Reviewer #1 (Remarks to the Author):

The manuscript by Alejo and co-workers demonstrates the importance of the dual TNF/chemokine decoy receptor CrmD, secreted by ectromelia virus, as an essential virulence factor in the progression of mousepox. The authors use recombinant viruses with a deletion of the CrmD encoding gene, or expressing constructs covering the N-terminal TNF binding (CRD), or C-terminal chemokine binding (SECRET) domain of CrmD.

The results presented in the study show that both the TNF and chemokine binding domains of CrmD are necessary for CrmD activity *in vivo*, since only the CrmD deleted virus, and not the CRD or SECRET domain expressing virus, shows a strongly attenuated phenotype. Although initially viral replication does not seem to be hindered in the CrmD deleted virus, lower titers are found in the spleen and liver during later stages of infection. The authors observe that the presence of the SECRET domain of CrmD is especially important for virus replication in the liver, and that reduced damage to the liver is most likely the cause of survival of mice infected with the CrmD deleted virus.

Furthermore, by using the different virus strains, the authors were able to dissect the contribution of the TNF binding and chemokine binding domains of CrmD during infection. The TNF binding CRD domain is shown to impair early NK cell activation, while both the TNF and chemokine binding (SECRET) domains are necessary for the inhibition of CD8 T cells in the spleen. Finally, the authors use recombinantly expressed CrmD to immunize mice, and show that almost 70% of CrmD immunized population was able to survive lethal doses of ectromelia virus. These results open opportunities for designing new protein based vaccines to prevent/treat human poxviral infections using recombinant CrmB from variola virus, the causative agent of human smallpox.

The manuscript is without doubt very interesting within the fields of immunology/virology/decoy receptors. The paper is clearly written, and the presented data are convincing. However, as I am not an expert in immunohistochemistry or flow-cytometry, I leave it to the other reviewers to assess the technical soundness of these conducted experiments and their statistical analysis. Although I do not have major remarks, I'd like to propose some changes and additions which may improve the manuscript:

- In the introduction, the authors describe that the SECRET domain adopts a fold common to other poxvirus encoded viral chemokine binding proteins. The fold of the SECRET domain is found in poxvirus encoded proteins with functionalities that go beyond sequestration of chemokines: e.g. a soluble MHC class I-binding protein from cowpox virus (CPXV203, McCoy et al., 2012) and a GM-CSF and IL-2 binding protein from the parapox orf virus (GIF, Felix et al., 2016). While the term 'SECRET' domain (smallpox virus-encoded chemokine receptor) was proposed in a paper of the first author

(Alejo et al, PNAS, 2006) this domain was recently renamed to PIE domain (poxviral immune evasion, Nelson et al. 2015). In my opinion this term better describes the versatility of this structural domain in viral immunomodulation. While this is not important for the main conclusions presented in the paper, the authors might mention this either in the introduction, or in the discussion of the manuscript.

- Some of the main conclusions and a part of the discussion is based on binding studies (SPR) between CrmD and a set of human and mouse chemokines (supplementary table in Alejo et al., 2006). There seem to be some discrepancies between this table and the main text of the manuscript:

page 11 (results): "CCL28 and CCL27, both of which are bound with high affinity by the CrmD SECRET domain". CCL27 is not present in the supplementary table from Alejo et al., 2006. The table shows affinities for: hCCL28, mCCL25, hCCL20, hCXCL12b, hCXCL13 and hCXCL14.

page 18 (discussion): "the SECRET domain binds CXCL11, one of the ligands of the chemokine receptor CXCR3": CXCL11 is not present in the supplementary table from Alejo et al., 2006.

- The SPR binding studies were presented in the 2006 paper as a supplementary table without showing any binding curves, most probably because CrmD was not the main topic of this paper, which mainly described variola virus CrmB. Since the authors have already conducted the SPR experiments, I would find it informative to include the CrmD-chemokine binding data (table with affinities as well as binding curves) in the supplementary, or even main section of the manuscript.

- Small textual issues:

- page 14: use correct reference to paper of LJ Sigal.

- page 15: "70% of the CrmD immunized mice survived infection.": 5 out of 15 mice succumbed to infection, change 70% to 67%

- i.n. and s.c.: define abbreviations on page 2.

- Page 37: space lacking between "or" and "ECTVRevCRD" in the description of Figure 1.

Reviewer #2 (Remarks to the Author):

The authors provide an interesting approach to evaluating the role of CrmD in ECTV-induced virulence in mice. The approach provides mechanistic as well as efficacy data. There are several items that need to be addressed before this reviewer considers this manuscript acceptable for publication.

1) Although a negative control group consisting of animals immunized with PBS and challenged with ECTV was used for the efficacy arm of the study, it does not appear that a PBS challenge groups was used as a control group for the mechanistic studies. Such a control is critical to evaluating the mechanistic endpoints. These data need to be included.

2) Regarding the virus used for infection, was the virus stock certified to be free of contamination, endotoxin, and Mycoplasma? This needs to be addressed.

3) There is minimal information on statistical analyses used. There is one mention of a Student's t-test on percentages in the Methods (line 595). However, for several figures, there is mention of statistical significance with no description of methods. Moreover, for statistical analyses performed on data in which time is a variable, it is highly suggested that an ANOVA with post-hoc analysis and adjustments made (e.g., Bonferroni) for error associated multiple comparisons. It is recommended that the authors provide the necessary level of detail and specifics regarding statistical analysis and re-run such analyses as appropriate.

4) For the efficacy study, please provide the rationale for using 1000 pfu. Also, please provide the actual dose as the 1000 is theoretical.

5) For IHC, it is recommended that the authors provide micrographs of tissue samples from negative control mice as well as stained sections in which the primary antibody has been eliminated and/or the use of an isotype control. These control sections are needed in order for this reviewer to assess non-specific background.

6) In Table 1, the dose administered (pfu) is theoretical. Please provide the actual dose that was given. Also, please provide ranges for all MTD numbers (Table 1).

Remove reference to unpublished data (lines 470-471). Also, please provide/show all data that are mentioned as "no shown" in the text.

7) In the Methods, please provide the antibodies/staining panel used for flow cytometric analysis.

8) The data in Figure 7A are interesting and need to be explained further in the corresponding results section. The results state that sera from 14 of 15 CrmD-immunized mice were shown to neutralize CrmD in vitro. From the figure, it appears that 15 of 15 exhibited some type of neutralization. Moreover, the authors need to provide information as to what %cell viability is considered to be a threshold for neutralization of CrmD that indicates a protective factor, such as X-fold above TNF alone.

Reviewer #3 (Remarks to the Author):

This is an interesting and well-written manuscript that studies the ectromelia gene *crmd* and its role in virulence and mousepox pathogenesis, and that presents results that are of interest for virus-host interactions and immune responses so relevant for a wider audience. The model employed and the data presented are sound and, in some cases, astonishing: a 6-log reduction in LD50 in the absence of CrmD is pretty remarkable! CrmD is then a virulence factor and has (at least) 2 functions: anti-TNF and anti-chemokine activity. The authors argue that chemokines contribute to the mouse response to pox infection only in the presence of anti-TNF activity (e.g. lines 291-295, 472-474). This conclusion essentially derives from the fact that a virus deficient for anti-TNF activity (point mutant N77F) was avirulent like the virus lacking CrmD, but a virus proficient for anti-TNF activity only (expressing CRD) did not rescue full virulence. To this reviewer, there are several points to address to be able to draw that conclusion.

1) Could the N77F mutant be simply 'dead' for anti-chemokine (as well as anti-TNF) activity? It is mentioned in the text that the full characterisation of this mutant is to be published, but perhaps some data could be added in this manuscript to illustrate the ability of this mutant to block chemokines to the same extent as wild-type CrmD. Indeed, the expression levels in Fig. 5B seem to indicate that the N77F mutant virus (labelled revSECRET) expresses lower levels of CrmD compared to wild-type, which could mask the real levels of virulence associated to this protein.

2) Assuming that the N77F mutant virus retains an unaltered anti-chemokine activity, the data presented here indicates that the anti-TNF activity of CrmD is responsible for its contribution to virulence. The observation that a virus expressing CRD (so the anti-TNF domain) does not recover full virulence could be explained by the fact that (i) other parts of the molecule could have other (yet unknown) functions and/or (ii) the expression/stability of the CRD-only CrmD in vivo may not be equivalent to that of wild-type. I understand that addressing the former may seem beyond the scope of this manuscript (although the possibility exists and should be mentioned somewhere in the

discussion), but the latter can be addressed by immunoblotting infected animal tissues with the available anti-CrmD antibody. As indicated by the authors 'viral replication was not hindered at the site of inoculation' (lines 238-240) and Fig. 4 shows similar titers for all viruses (including 'revCRD') at 3 dpi in spleen and liver. Any of these samples should reveal the levels of expression of CRD at a time where the host immune response has not affected viral replication.

3) Assuming the above pans out, a critical mechanistic insight into why the CRD-only virus is attenuated compared to the wild-type is its inability to cause lymphopenia in spleens at 7 dpi despite inhibiting NK cell activation in the lymph node at 3 dpi. This observation is very interesting and should be substantiated with the analysis of pro-inflammatory cytokine and chemokine transcription levels (including type I interferon and NF- κ B signalling and maybe also CXCL11 [as suggested by the authors in the discussion]) at a time where viral loads are similar between viruses (3 dpi?). Such data will complete the picture as for how CrmD modulates host anti-viral immunity.

Other points:

1. CrmD is diploid. Although clearly mentioned for the revertant virus, it is not clear as one reads whether the CRD revertant and N77F mutant viruses also have 2 gene copies reinserted. I am aware this is clarified in the M&M, but should also be mentioned in the main text.
2. Line 292-293: 'virulence factor in vivo' is redundant. Remove 'in vivo'.
3. Page 14: this section is explained in a manner that does not correlate with the legend for Fig. 6. I believe the legend is incorrect and should be modified.
4. Line 318-319: ??
5. Line 511: should read 'Modified vaccinia virus Ankara'.
6. The section M&M seems to lack detail throughout with the exception of the construction of recombinant viruses and IHC analyses, which are extended in the supplementary material. A clear example is the expression of protein CrmD in a baculovirus system; this needs to be explained or referred to.
7. Fig 7, panel A: the crosses indicating mice that succumbed to lethal challenge are confusing (sera from immunised mice are tested at this point). Is it really informative? Also, could this data be presented in a different way so statistical analysis can be performed?

Reviewer #4 (Remarks to the Author):

Ectromelia virus (ECTV) encoded cytokine response modifier D (CrmD) contains a tumor necrosis factor (TNF) binding domain (CRD) and the smallpox virus-encoded chemokine receptor (SECRET)

domain. It has not been clear how the 2 domains of the CrmD protein contribute to subversion or evasion of the host immune response.

Through the use of a number of mutant and revertant viruses Alejo and colleagues show that virus-encoded CrmD reduces inflammation and as a result the effectiveness of the host antiviral response is also reduced. There was minimal inflammation (footpad swelling) in infected mice when CrmD was intact whereas in its absence, inflammation was augmented. Deletion of CrmD also resulted in significant attenuation of the virus in vivo. The normally susceptible BALB/c mice fully recovered from infection with high doses of the virus. The authors also show that both domains, i.e. CRD and SECRET, co-operate and are required for the full expression of CrmD activity. Their data is consistent with a role for both TNF and specific chemokines in providing protection against lethal ECTV infection.

The authors have previously suggested that viral TNFRs like CrmD could be used as additional therapeutics to treat diseases in which TNF is known cause significant damage and pathology. Based on results presented in the current manuscript, they suggest that that combining a chemokine inhibitory domain to the human soluble TNFR therapeutics may increase their clinical efficacy in certain settings.

Finally, the authors have found that vaccination of mice with recombinant CrmD protein conferred protection against lethal ECTV infection.

This is strong study and the results are convincing and importantly novel. It indicates, perhaps for the first time, that 2 virally encoded proteins can collaborate to exhibit the maximal effects on modulating the host response. There are nonetheless a number of concerns/comments that need to be clarified.

Questions/Comments for the Authors

Table 2. It will important to provide representative H&E sections, particularly for the liver, so that the reader will be able to understand how the - (0 %); + slight (about 25 % necrosis); ++ moderate (about 50 % necrosis); +++ very intense (about 90-100 % necrosis) degrees of necrosis were semi-quantified. This could be presented as supplementary data.

Figure 1.

The single-step growth curves of viruses were undertaken in BSC-1 cells (African green monkey cells). Do the authors know if CrmD interacts with African green monkey TNF or lymphotoxin? Would the results be different if the cells used were of murine origin? The concern here is that there is a possibility that if murine cells had been used, the authors may have seen a difference in replication rates between the various mutants through interaction of CrmD with murine TNF.

histology

Figure 2.

The % initial weight panels. Top 2 panels show data with filled squares. What are these. Mock infected animals?

Survival – Explain why ECTVRevCrmD appears more virulent than ECTV at the 10 PFU dose. Is there a significant difference?? Are there any statistically significant differences between ECTVRevCRD and ECTVdeltaCrmD at 1000 PFU?

It will be useful to include in the legend to depict the different viruses like you have done in Figure 5.

Figure 3

The panels (C-H) should be made larger as the details are not clear. Panel H: Which are the CCR10 positive cells? The blue or red cells? Without the staining of a section from ECTV infected mice, it will be impossible to tell if there are any differences between the 2 viruses.

Figure 4. Histological sections are not clear. It is impossible to assess details in the insets in L, M and N even if magnified. Authors will have to provide better quality figures.

Figure 6. Panels A and B provide % NK cells whereas panels C and D provide numbers of CD3+CD8+ T lymphocytes. It would be useful to have numbers for both cell types. Are there differences in numbers of NK cells?

Representative flow cytometry plots for NK cells and CD8+ T lymphocytes should be included as supplementary data.

Discussion

The authors suggest that secreted poxviral immunomodulators, used singly or in combination with other proteins, may be effective subunit vaccines in preventing lethal poxvirus infections. They also point out that smallpox VACV vaccines Dryvax and Modified virus Ankara do not express the CrmB protein and will not induce a neutralizing response against CrmB expressed by VARV or MPXV. It is not clear what the intent of that statement is and should be clarified. The authors would recognize that Dryvax was very effective in eradicating smallpox, arguing against a need for antibodies against CrmD for protection against VARV.

The authors would also be aware that antibodies generated by vaccination against envelope proteins (even if in the form of subunit vaccines) would bind to virus and mediate effector functions even before the virus particle can enter a host cell. Such antibodies would be expected to be more effective than antibodies against secreted viral immuno-modulator – these are only produced once a host cell has been infected and virus replication must occur. It is important that the authors consider these differences and provide a balanced discussion.

The clinical use of TNFRs to treat various inflammatory conditions has shown that reactivation of a number of specific pathogens can occur in treated individuals. In their discussion, the authors should also discuss how combining a chemokine inhibitory domain to the TNFR therapeutics could potentially broaden the types of pathogens that may be reactivated or cause more serious disease.

Other comments:

Page 14, 3rd para, 3rd line – references not cited. Authors should be aware that lymphopenia in mousepox has been described several decades ago by Mims so those references should also be included and acknowledged.

There are several typographical errors throughout the manuscript.

RESPONSE TO REVIEWERS' COMMENTS

We have addressed the reviewer's comments and have added more data to support the manuscript:

- *A new Fig. 4 showing the interaction of CrmD with mouse chemokines and their affinity constants.*
- *A new Fig. 6 describing the properties of the N77F mutant for of CrmD: binding and biological activity for TNF and chemokines.*
- *A new Supplementary Fig. 1 showing controls of immunohistochemistry.*
- *A new Supplementary Fig. 2 illustrating the different levels of liver and spleen necrosis.*
- *A new Fig. 8 describing in more detail the flow cytometry analysis of immune cells.*
- *The statistical analysis has been performed using multiple t tests with false discovery rate $Q=1\%$ and ANOVA.*

Reviewer #1 (Remarks to the Author):

The manuscript by Alejo and co-workers demonstrates the importance of the dual TNF/chemokine decoy receptor CrmD, secreted by ectromelia virus, as an essential virulence factor in the progression of mousepox. The authors use recombinant viruses with a deletion of the CrmD encoding gene, or expressing constructs covering the N-terminal TNF binding (CRD), or C-terminal chemokine binding (SECRET) domain of CrmD.

The results presented in the study show that both the TNF and chemokine binding domains of CrmD are necessary for CrmD activity in vivo, since only the CrmD deleted virus, and not the CRD or SECRET domain expressing virus, shows a strongly attenuated phenotype. Although initially viral replication does not seem to be hindered in the CrmD deleted virus, lower titers are found in the spleen and liver during later stages of infection. The authors observe that the presence of the SECRET domain of CrmD is especially important for virus replication in the liver, and that reduced damage to the liver is most likely the cause of survival of mice infected with the CrmD deleted virus.

Furthermore, by using the different virus strains, the authors were able to dissect the contribution of the TNF binding and chemokine binding domains of CrmD during infection. The TNF binding CRD domain is shown to impair early NK cell activation, while both the TNF and chemokine binding (SECRET) domains are necessary for the inhibition of CD8 T cells in the spleen. Finally, the authors use recombinantly expressed CrmD to immunize mice, and show that almost 70% of CrmD immunized population was able to survive lethal doses of ectromelia virus. These results open opportunities for designing new protein based vaccines to prevent/treat human poxviral infections using recombinant CrmB from variola virus, the causative agent of human smallpox.

The manuscript is without doubt very interesting within the fields of immunology/virology/decoy receptors. The paper is clearly written, and the presented data are convincing. However, as I am not an expert in immunohistochemistry or flow-cytometry, I leave it to the other reviewers to assess the technical soundness of these conducted experiments and their statistical analysis. Although I do not have major remarks, I'd like to propose some changes and additions which may improve the manuscript:

- In the introduction, the authors describe that the SECRET domain adopts a fold common to other poxvirus encoded viral chemokine binding proteins. The fold of the SECRET domain is found in poxvirus encoded proteins with functionalities that go beyond sequestration of chemokines: e.g. a soluble MHC class I-binding protein from cowpox virus (CPXV203, McCoy et al., 2012) and a GM-CSF and IL-2 binding protein from the parapox orf virus (GIF, Felix et al., 2016). While the term 'SECRET' domain (smallpox virus-encoded chemokine receptor) was proposed in a paper of the first author (Alejo et al, PNAS, 2006) this domain was recently renamed to PIE domain (poxviral immune evasion, Nelson et al. 2015). In my opinion this term better describes the versatility of this structural domain in viral immunomodulation. While this is not important for the main conclusions presented in the paper, the authors might mention this either in the introduction, or in the discussion of the manuscript.

RESPONSE: We thank the reviewer for the suggestion, and we agree that we should mention the poxvirus immune evasion (PIE) domain (Nelson et al. 2015). However, SECRET refers to a subset of PIE domains that bind chemokines and may be fused to TNF binding domains. The SECRET domain does not include the 35 kDa nor the A41 chemokine binding proteins, although these proteins have a similar structural folding and bind chemokines. Because of these reasons we prefer to keep the name 'SECRET' to refer to a specific group of domains that bind chemokines. We have modified the Introduction to add references to the CPXV203 and GIF proteins (McCoy et al. 2012; Felix et al. 2016), and the multifunctional PIE domain (Nelson et al. 2015) (lines 64 and 70-77). We also added a relevant recent review on soluble decoy receptors by Felix and Savvides, 2017).

- Some of the main conclusions and a part of the discussion is based on binding studies (SPR) between CrmD and a set of human and mouse chemokines (supplementary table in Alejo et al., 2006). There seem to be some discrepancies between this table and the main text of the manuscript:

page 11 (results): "CCL28 and CCL27, both of which are bound with high affinity by the CrmD SECRET domain". CCL27 is not present in the supplementary table from Alejo et al., 2006. The table shows affinities for: hCCL28, mCCL25, hCCL20, hCXCL12b, hCXCL13 and hCXCL14.

page 18 (discussion): "the SECRET domain binds CXCL11, one of the ligands of the chemokine receptor CXCR3": CXCL11 is not present in the supplementary table from Alejo et al., 2006.

RESPONSE:

We thank the reviewer for pointing out these discrepancies. Our previous publication by Alejo et al. 2006 included binding studies of CrmD to human chemokines. We realized now that mouse chemokines were not included in the previous study. We have now added binding studies of CrmD to mouse chemokines and have included in the present submission the data, shown in a new Fig.4. This Figure includes: (A) results of the binding (SPR) studies with mouse chemokines, showing binding to 11 mouse chemokines; (B) binding affinity studies to those mouse chemokines binding with higher affinity; and (C) representative kinetic studies used to determine the affinities of mouse Cxcl13 and Ccl25.

In response to reviewer's comments: CrmD binds mouse CCL27 and CCL28, but CCL27 with higher affinity; and CrmD binds mouse CXCL11 with high affinity (see new Fig.4).

These experiments are now included in the text (lines 223-224 and 229-232) and new Fig.4. The methodology for CrmD expression, binding studies and chemotaxis assays is now described in Methods (lines 564-573 and 582-610).

- The SPR binding studies were presented in the 2006 paper as a supplementary table without showing any binding curves, most probably because CrmD was not the main topic of this paper, which mainly described variola virus CrmB. Since the authors have already conducted the SPR experiments, I would find it informative to include the CrmD-chemokine binding data (table with affinities as well as binding curves) in the supplementary, or even main section of the manuscript.

RESPONSE: As described above, binding studies to mouse chemokines are now included in the text and new Fig. 4, showing representative examples of binding curves.

- Small textual issues:
- page 14: use correct reference to paper of LJ Sigal.

RESPONSE: We apologize for the mistake. We have now added the two missing references (Fang et al. 2006; Fang et al. 2008) and also included an earlier observation by C. Mims (1964), as suggested by reviewer 4. See lines 335-336.

- page 15: "70% of the CrmD immunized mice survived infection.": 5 out of 15 mice succumbed to infection, change 70% to 67%

RESPONSE: *We have modified the text as suggested (line 360).*

- i.n. and s.c.: define abbreviations on page 2.

RESPONSE: *The abbreviations are now defined on page 2.*

- Page 37: space lacking between “or” and “ECTVRevCRD” in the description of Figure 1.

RESPONSE: *We have modified the text as suggested.*

Reviewer #2 (Remarks to the Author):

The authors provide an interesting approach to evaluating the role of CrmD in ECTV-induced virulence in mice. The approach provides mechanistic as well as efficacy data. There are several items that need to be addressed before this reviewer considers this manuscript acceptable for publication.

1) Although a negative control group consisting of animals immunized with PBS and challenged with ECTV was used for the efficacy arm of the study, it does not appear that a PBS challenge groups was used as a control group for the mechanistic studies. Such a control is critical to evaluating the mechanistic endpoints. These data need to be included.

RESPONSE: *We agree with the reviewer’s concern. We can confirm that our control ‘mock-infected’ mice were actually ‘PBS-inoculated’, and we apologize for not including this information. The text has been modified to make this clear: in the Results section ‘Modulation of NK cell and CD8 T cell responses by CrmD’ (lines 313 and 318); Methods ‘Flow cytometry’ (line 671) and Fig.8 (line 1129).*

2) Regarding the virus used for infection, was the virus stock certified to be free of contamination, endotoxin, and Mycoplasma? This needs to be addressed.

RESPONSE: *The requested information has been added to the Methods section ‘Cells and viruses’ (lines 560-562): Viral stocks were routinely tested for the absence of mycoplasma and the endotoxin levels detected using ToxiSensor Chromogenic LAL Endotoxin Assay kit (GenScript) were under 0.3 EU/ml.*

3) There is minimal information on statistical analyses used. There is one mention of a Student’s t-test on percentages in the Methods (line 595). However, for several figures, there is mention of statistical significance with no description of methods. Moreover, for statistical analyses performed on data in which time is a variable, it is highly suggested that an ANOVA with post-hoc analysis and adjustments made (e.g., Bonferroni) for error associated multiple comparisons. It is recommended that the authors provide the necessary level of detail and specifics regarding statistical analysis and re-run such analyses as appropriate.

RESPONSE:

The statistical analysis has been performed again following the suggestions of the reviewer, and the new version of the manuscript has incorporated the new analyses. This new statistical analyses confirmed significant differences similar to those described in our initial statistical analysis.

1. Data presented in new Fig. 2 and new Fig. 7e. Multiple t tests with false discovery rate $Q=1\%$. Analyses were performed up to times post infection at which survival rates in the corresponding group were above 50% (In Fig. 2, 10 dpi for ECTVRevCrmD infected animals at the 10 pfu dose and 9 dpi for all other ECTV and ECTVRevCrmD infected groups; up to end of experiment for ECTV RevCrmD CRD and ECTV DelCrmD infected groups) (In Fig. 7e, 7 dpi for ECTV RevCrmD and up to the end of the experiment for ECTV RevCRD, ECTV DelCrmD and ECTV RevSECRET).

2. Data presented in new Fig. 7f and new Fig. 8b-8f. ANOVA analyses with Bonferroni multiple comparison tests.

ANOVA analyses could be performed in some experiments, as indicated, for comparisons among groups and times post-infection at which no mortalities were observed. This is the reason why ANOVA could not

be performed for all experiments. In the case of the animal experiment described in new Fig.2, ANOVA analysis was performed on the first days (before mortalities were observed) and confirmed statistically significant time and group effects.

The new statistical analysis has been described in the Methods section (lines 631-638) and the legends to new Figures 2 (lines 1030-1032), 7 (lines 1117 and 1122-1124) and 8 (lines 1143-1144).

The *p* values obtained with multiple *t* tests with false discovery rate $Q=1\%$ in the experiments described are listed below:

New Fig. 2. 10 pfu, % initial weight:

ECTV vs ECTV DelCrmD: 7dpi: $p=0,000131676$; 8dpi: $p=0,00010492$; 9dpi: $p=0,000497776$

ECTVRevCrmD vs ECTV DelCrmD: 8dpi: $p=2,814097e-005$; 9dpi: $p=0,000899878$; 10dpi: $p=0,000709553$

New Fig. 2. 10 pfu, footpad swelling:

ECTV vs ECTV DelCrmD: 9dpi: $p=1,114101e-007$

ECTVRevCrmD vs ECTV DelCrmD: 9dpi: $p=1,497983e-006$; 10dpi: $p=2,937252e-011$

DelCrmD vs CRD: 9dpi: $p=2,499640e-008$; 10dpi: $p=1,148922e-007$; 13dpi: $p=0,000834105$

New Fig. 2. 1000 pfu, %initial weight:

ECTV vs ECTV DelCrmD: 7dpi: $p=0,000284872$; 8dpi: $p=0,000293363$; 9dpi: $p=0,000370366$;

ECTVRevCrmD vs ECTV DelCrmD: 8dpi: $p=0,000135491$;

New Fig. 2. 1000 pfu, footpad swelling:

ECTV vs ECTV DelCrmD: 7dpi: $p=1,291826e-006$; 8dpi: $p=5,476546e-019$; 9dpi: $p=1,217545e-017$

ECTVRevCrmD vs ECTV DelCrmD: 7dpi: $p=1,574303e-006$; 8dpi: $p=1,735191e-018$; 9dpi: $p=3,394428e-017$

DelCrmD vs CRD: 8dpi: $p=1,294794e-008$; 9dpi: $p=0,000131255$

ECTVRevCrmD vs CRD: 8dpi: $p=0,00282739$; 9dpi: $p=4,632347e-008$

ECTV vs CRD: 9dpi: $p=1,139449e-007$;

New Fig. 7e footpad swelling:

DelCrmD vs RevCrmD: 5dpi, $p=0.00095$; 6dpi, $p=0.000032$; 7dpi, $p=0.000028$

DelCrmD vs RevCRD: 5dpi, $p=0.00095$; 6dpi, $p=0.000032$; 7dpi, $p=0.000028$; 8dpi, $p=0.00016$; 9dpi, $p=0.00137$; 10dpi, $p=0.00256$.

DelCrmD vs RevSECRET: 5dpi, $p=0.00095$; 6dpi, $p=0.0148$; 7dpi, $p=0.00202$; 8dpi, $p=0.00046$; 9dpi, $p=0.0623$; 10dpi, $p=0.00382$.

RevCRD vs RevSECRET: 6dpi, $p=0.04487$; 7dpi, $p=0.01707$; 8dpi, $p=0.00796$; 9dpi, $p=0.04357$; 10 dpi, $p=0.0665$.

RevCrmD vs RevSECRET: 6dpi, $p=0.04487$; 7dpi, $p=0.01707$

4) For the efficacy study, please provide the rationale for using 1000 pfu. Also, please provide the actual dose as the 1000 is theoretical.

RESPONSE:

To test the protective response induced after immunization with purified CrmD we infected mice with a dose of 1000 pfu, corresponding to approx. 100-fold LD50. We wanted to challenge mice with a high dose to test the induction of a potent protective response. We have included a sentence to explain the rationale (lines 348-350).

We always titrate the amount of virus present in the inoculum used to infect mice, to confirm the dose. If we get a virus titre similar (within a 10% error) to that we expected from previous titrations we consider the virus titer is correct and no mistakes were made when diluting the virus stocks. We confirm that the titrations of the viruses on the same day used for the experiments shown in the manuscript were within the 10% error range. A sentence has been added in the Methods section (lines 625-627).

5) For IHC, it is recommended that the authors provide micrographs of tissue samples from negative control mice as well as stained sections in which the primary antibody has been eliminated and/or the use of an isotype control. These control sections are needed in order for this reviewer to assess non-specific background.

RESPONSE: *We have now included a new Supplementary Figure 1 where we show the negative staining of the anti-virus and anti-CrmD antibodies of tissues from uninfected mice, and an explanation in the text (lines 243-245).*

6) In Table 1, the dose administered (pfu) is theoretical. Please provide the actual dose that was given. Also, please provide ranges for all MTD numbers (Table 1).

RESPONSE: *As indicated in above (response No.4) we titrated the virus inoculum, but keep the theoretical titre if the result is within a 10% error. Ranges for all MTD numbers are now included in Table 1.*

Remove reference to unpublished data (lines 470-471). Also, please provide/show all data that are mentioned as “no shown” in the text.

RESPONSE: *The reference to unpublished data has been removed, and we have checked that no reference to ‘not shown’ remains in the text. Reference to ‘anti-virus and anti-CrmD staining of sections from footpads of infected mice at 3 dpi’ has been removed (line 242) and we only refer to 7 dpi (shown in new Fig.5 and Supplementary Fig.1).*

7) In the Methods, please provide the antibodies/staining panel used for flow cytometric analysis.

RESPONSE: *Information on antibodies used for flow cytometry is now included in Methods (lines 676-683).*

8) The data in Figure 7A are interesting and need to be explained further in the corresponding results section. The results state that sera from 14 of 15 CrmD-immunized mice were shown to neutralize CrmD in vitro. From the figure, it appears that 15 of 15 exhibited some type of neutralization. Moreover, the authors need to provide information as to what %cell viability is considered to be a threshold for neutralization of CrmD that indicates a protective factor, such as X-fold above TNF alone.

RESPONSE: *We have defined in the text a threshold of 50% cell viability for CrmD neutralization (line 352). The bars in Fig. 7a (new Fig. 9a) represent the neutralization ability of 1 or 2 µl of mouse serum. All sera from CrmD immunized mice showed CrmD neutralization activity (less than 50% cell viability) at 1 µl or 2µl. Serum CrmD 6 was the only one that did not neutralize, since both doses are above 100% cell viability. We hope this will be clear with the modified sentence.*

Reviewer #3 (Remarks to the Author):

This is an interesting and well-written manuscript that studies the ectromelia gene *crmd* and its role in virulence and mousepox pathogenesis, and that presents results that are of interest for virus-host interactions and immune responses so relevant for a wider audience. The model employed and the data presented are sound and, in some cases, astonishing: a 6-log reduction in LD50 in the absence of *Crmd* is pretty remarkable! *Crmd* is then a virulence factor and has (at least) 2 functions: anti-TNF and anti-chemokine activity. The authors argue that chemokines contribute to the mouse response to pox infection only in the presence of anti-TNF activity (e.g. lines 291-295, 472-474). This conclusion essentially derives from the fact that a virus deficient for anti-TNF activity (point mutant N77F) was avirulent like the virus lacking *Crmd*, but a virus proficient for anti-TNF activity only (expressing CRD) did not rescue full virulence. To this reviewer, there are several points to address to be able to draw that conclusion.

1) Could the N77F mutant be simply 'dead' for anti-chemokine (as well as anti-TNF) activity? It is mentioned in the text that the full characterisation of this mutant is to be published, but perhaps some data could be added in this manuscript to illustrate the ability of this mutant to block chemokines to the same extent as wild-type *Crmd*. Indeed, the expression levels in Fig. 5B seem to indicate that the N77F mutant virus (labelled revSECRET) expresses lower levels of *Crmd* compared to wild-type, which could mask the real levels of virulence associated to this protein.

RESPONSE:

We agree with the reviewer that this information is relevant for this manuscript. We have prepared a new Figure 6 showing that purification of the CrmD N77F mutant, its binding properties (TNF and chemokine) and anti-TNF and anti-chemokine biological properties, compared to the wild type CrmD protein. The results show that the CrmD N77F mutant has lost the ability to bind and inhibit TNF, but retains the capacity of binding Ccl25 and blocking Ccl25-induced cell migration, with similar potency as wild type CrmD.

We have included a new Figure 6, incorporated text in the Results section (lines 274-278) and described the experiments in Methods (lines 564-573, 582-591 and 593-610).

Regarding the expression level of the CrmD N77F mutant from ECTV RevSECRET we agree that the protein expression appears to be lower, but the loading of this samples was lower, as shown from the intensity of the vCKBP band used as a control for loading. Densitometry of the CrmD and vCKBP bands in lanes 3, 4 and 5 (new Fig. 7b) show a similar relative density when considering the intensity of vCKBP: wild type CrmD (1), CRD (0,88) and CrmD N77F mutant (1,25). We have included a short comment in the text (line 283).

	CrmD			vCKBP			Adjusted Relative density
	Area	Percent	Relative density	Area	Percent	Relative density	
Lane 3	158788,123	37,109	1	5808,154	37,128	1	1
Lane 4	166563,063	38,926	1,048963863	6851,154	43,795	1,179567981	0,889278007
Lane 5	102548,395	23,966	0,645827158	2984,326	19,077	0,513817065	1,256920413

2) Assuming that the N77F mutant virus retains an unaltered anti-chemokine activity, the data presented here indicates that the anti-TNF activity of *Crmd* is responsible for its contribution to virulence. The observation that a virus expressing CRD (so the anti-TNF domain) does not recover full virulence could be explained by the fact that (i) other parts of the molecule could have other (yet unknown) functions and/or (ii) the expression/stability of the CRD-only *Crmd* in vivo may not be equivalent to that of wild-type. I understand that addressing the former may seem beyond the scope of this manuscript (although the possibility exists and should be mentioned somewhere in the discussion), but the latter can be addressed by immunoblotting infected animal tissues with the available anti-*Crmd* antibody. As indicated by the authors 'viral replication was not hindered at the site of inoculation' (lines 238-240) and Fig. 4 shows similar titers for all viruses (including 'revCRD') at 3 dpi in spleen and liver. Any of these samples should reveal the levels of expression of CRD at a time where the host immune response has not affected viral replication.

RESPONSE:

Regarding the attenuation of the virus expressing CRD (anti-TNF activity) we agree with the reviewer that it could be explained by the lack of anti-chemokine activity or other activities not identified in the SECRET domain. We think that the most relevant point here is that expression of anti-TNF activity is not sufficient to confer virulence. To avoid a complex discussion, we prefer not to raise the possibility of unknown activities (an argument that could be applied to any protein characterized in animal models) and to focus on the anti-chemokine activity of the SECRET domain.

The expression of CRD from ECTV-infected cultures is equivalent to that of full-length CrmD, and most important, the anti-TNF activity is comparable (Fig. 1 and Fig. 7). We agree that the stability of the protein may be different, although we think that this is unlikely considering that the post-translational modifications such as glycosylation should be similar. Testing the stability of the protein in tissue samples where virus replication is similar has been suggested by the reviewer. Figure 5c,d,e shows the same level of viral replication of ECTV RevCrmD, Δ CrmD and RevCRD in the footpad. Staining with anti-CrmD antibodies of these samples (Figure 5f,g,h) shows a similar signal in ECTV RevCrmD and ECTV RevCRD, suggesting comparable stability of CRD in vivo. A short reference to this point has been added to the text (line 243).

3) Assuming the above pans out, a critical mechanistic insight into why the CRD-only virus is attenuated compared to the wild-type is its inability to cause lymphopenia in spleens at 7 dpi despite inhibiting NK cell activation in the lymph node at 3 dpi. This observation is very interesting and should be substantiated with the analysis of pro-inflammatory cytokine and chemokine transcription levels (including type I interferon and NF- κ B signalling and maybe also CXCL11 [as suggested by the authors in the discussion]) at a time where viral loads are similar between viruses (3 dpi?). Such data will complete the picture as for how CrmD modulates host anti-viral immunity.

RESPONSE: We agree with the reviewer's suggestion that analysis of the expression of cytokines and chemokines will be very informative to understand the immunomodulatory activities of CrmD. In fact, this is what we are doing, but we are following a global approach characterizing the host transcriptome in different organs of mice infected with the collection of CrmD virus mutants described here. The RNAseq analysis of the host transcriptome will be more informative than looking to a limited number of genes, but this is a complex project and will be the subject of an independent study in which we hope to identify specific immune pathways controlled by CrmD.

Other points:

1. CrmD is diploid. Although clearly mentioned for the revertant virus, it is not clear as one reads whether the CRD revertant and N77F mutant viruses also have 2 gene copies reinserted. I am aware this is clarified in the M&M, but should also be mentioned in the main text.

RESPONSE: To clarify this point, we added 'the correct incorporation of two copies of the truncated CrmD TNF binding domain' in the construction of the CRD revertant' (lines 150-151). We have also added 'The complete genome sequence of this virus ... confirmed the incorporation of two copies of the CrmD N77F mutant gene' (lines 278-280).

2. Line 292-293: 'virulence factor in vivo' is redundant. Remove 'in vivo'.

RESPONSE: We have removed 'in vivo' as suggested.

3. Page 14: this section is explained in a manner that does not correlate with the legend for Fig. 6. I believe the legend is incorrect and should be modified.

RESPONSE: Figure 6 (new Figure 8) has been modified to address the comments from another reviewer, and both the legend and text (lines 318-341) have been modified.

4. Line 318-319: ??

RESPONSE: *We apologize for the mistake. We have now added the two missing references (Fang et al. 2006; Fang et al. 2008) and also included an earlier observation by C. Mims (1964), as suggested by reviewer 4.*

5. Line 511: should read 'Modified vaccinia virus Ankara'.

RESPONSE: *We have modified it as suggested.*

6. The section M&M seems to lack detail throughout with the exception of the construction of recombinant viruses and IHC analyses, which are extended in the supplementary material. A clear example is the expression of protein CrmD in a baculovirus system; this needs to be explained or referred to.

RESPONSE: *A new section in Methods describes the expression and purification of recombinant proteins (lines 564-573).*

7. Fig 7, panel A: the crosses indicating mice that succumbed to lethal challenge are confusing (sera from immunised mice are tested at this point). Is it really informative? Also, could this data be presented in a different way so statistical analysis can be performed?

RESPONSE: *We believe that marking the mice that succumbed to infection highlights the fact that many of the mice that died showed a weaker CrmD neutralization activity, as described in the text. We have not been able to find an alternative way to represent the data.*

Reviewer #4 (Remarks to the Author):

Ectromelia virus (ECTV) encoded cytokine response modifier D (CrmD) contains a tumor necrosis factor (TNF) binding domain (CRD) and the smallpox virus-encoded chemokine receptor (SECRET) domain. It has not been clear how the 2 domains of the CrmD protein contribute to subversion or evasion of the host immune response.

Through the use of a number of mutant and revertant viruses Alejo and colleagues show that virus-encoded CrmD reduces inflammation and as a result the effectiveness of the host antiviral response is also reduced. There was minimal inflammation (footpad swelling) in infected mice when CrmD was intact whereas in its absence, inflammation was augmented. Deletion of CrmD also resulted in significant attenuation of the virus in vivo. The normally susceptible BALB/c mice fully recovered from infection with high doses of the virus. The authors also show that both domains, i.e. CRD and SECRET, co-operate and are required for the full expression of CrmD activity. Their data is consistent with a role for both TNF and specific chemokines in providing protection against lethal ECTV infection.

The authors have previously suggested that viral TNFRs like CrmD could be used as additional therapeutics to treat diseases in which TNF is known cause significant damage and pathology. Based on results presented in the current manuscript, they suggest that that combining a chemokine inhibitory domain to the human soluble TNFR therapeutics may increase their clinical efficacy in certain settings.

Finally, the authors have found that vaccination of mice with recombinant CrmD protein conferred protection against lethal ECTV infection.

This is strong study and the results are convincing and importantly novel. It indicates, perhaps for the first time, that 2 virally encoded proteins can collaborate to exhibit the maximal effects on modulating the host response. There are nonetheless a number of concerns/comments that need to be clarified.

Questions/Comments for the Authors

Table 2. It will important to provide representative H&E sections, particularly for the liver, so that the reader will be able to understand how the - (0 %); + slight (about 25 % necrosis); ++ moderate (about 50

% necrosis); +++ very intense (about 90-100 % necrosis) degrees of necrosis were semi-quantified. This could be presented as supplementary data.

RESPONSE: *We added a new Supplementary Figure 2 showing representative examples of H&E sections that illustrate different degrees of necrosis, to complement the data presented in Table 2.*

Figure 1.

The single-step growth curves of viruses were undertaken in BSC-1 cells (African green monkey cells). Do the authors know if CrmD interacts with African green monkey TNF or lymphotoxin? Would the results be different if the cells used were of murine origin? The concern here is that there is a possibility that if murine cells had been used, the authors may have seen a difference in replication rates between the various mutants through interaction of CrmD with murine TNF.
histology

RESPONSE: *The main purpose of the growth curves shown in Figure 1d is to demonstrate that all viruses can replicate to wild type levels. Viral titrations in organs and immunostaining of tissues from infected mice also suggest that all viruses, including the CrmD deletion mutant, are competent for replication. The purpose of the study is to evaluate the effect of the TNF and chemokine inhibition on the anti-viral immune response in the infected mice. As the reviewer suggests, if TNF or lymphotoxin played a role during virus replication in tissue culture we would have observed limited virus replication in the CrmD deletion mutant. This was not observed in the culture system used. Although not tested formally, it is likely that CrmD interacts with TNF and lymphotoxin from monkeys. A previous study in VARV-infected monkeys suggests modulation of TNF by VARV, maybe through expression of CrmB, a protein with properties similar to CrmD (Rubins et al. 2004. PNAS USA 101:15190-15195). We do not know whether TNF may restrict viral replication in the cell culture system used, but even if TNF restricted the replication of the CrmD deletion mutant in a mouse cell system, it would not modify the conclusion of the study: that CrmD is dispensable for virus replication in cell culture but plays a major role in viral replication in infected mice.*

Figure 2.

The % initial weight panels. Top 2 panels show data with filled squares. What are these. Mock infected animals?

RESPONSE: *We apologize for not including this information. It corresponds to mock-infected mice inoculated with PBS, and this is now defined in the Figure legend.*

Survival – Explain why ECTVRevCrmD appears more virulent than ECTV at the 10 PFU dose. Is there a significant difference?? Are there any statistically significant differences between ECTVRevCRD and ECTVdeltaCrmD at 1000 PFU?

RESPONSE: *Survival curves were compared using the Logrank (Mantel-Cox) test. No significant differences are detected between ECTV and ECTVRevCrmD infected mice at either dose. No significant differences in terms of mortality between ECTVRevCRD and ECTVdeltaCrmD are detected either.*

It will be useful to include in the legend to depict the different viruses like you have done in Figure 5.

RESPONSE: *This has been changed in the legend to Figure 2.*

Figure 3

The panels (C-H) should be made larger as the details are not clear. Panel H: Which are the CCR10 positive cells? The blue or red cells? Without the staining of a section from ECTV infected mice, it will be impossible to tell if there are any differences between the 2 viruses.

RESPONSE: *We are submitting high quality Figures that will show the details of the sections better. We now indicate in the legend that positively labelled cells appear in red colour. Equivalent sections of mice infected with wild type ECTV show no infiltrate of immune cells, and hence no cell staining is observed in ECTV-infected tissues. This is now indicated in the legend (lines 1045-1047).*

Figure 4. Histological sections are not clear. It is impossible to assess details in the insets in L, M and N even if magnified. Authors will have to provide better quality figures.

RESPONSE: *We are submitting a high quality Figure.*

Figure 6. Panels A and B provide % NK cells whereas panels C and D provide numbers of CD3+CD8+ T lymphocytes. It would be useful to have numbers for both cell types. Are there differences in numbers of NK cells?

Representative flow cytometry plots for NK cells and CD8+ T lymphocytes should be included as supplementary data.

RESPONSE: *We include a new Figure 8 with more data as requested by the reviewer. The text in Methods and Results sections have also been expanded to provide a better explanation (lines 312-341 and 670-683).*

Discussion

The authors suggest that secreted poxviral immunomodulators, used singly or in combination with other proteins, may be effective subunit vaccines in preventing lethal poxvirus infections. They also point out that smallpox VACV vaccines Dryvax and Modified virus Ankara do not express the CrmB protein and will not induce a neutralizing response against CrmB expressed by VARV or MPXV. It is not clear what the intent of that statement is and should be clarified. The authors would recognize that Dryvax was very effective in eradicating smallpox, arguing against a need for antibodies against CrmD for protection against VARV.

RESPONSE: *We fully agree with the reviewer in that VACV Dryvax induced a protective response against VARV, by raising immunity against other viral proteins. We suggest that this response could be improved by inducing an anti-CrmB immune response, which is most relevant in a subunit vaccine. The Discussion has been modified to clarify this point (lines 531-532).*

The authors would also be aware that antibodies generated by vaccination against envelope proteins (even if in the form of subunit vaccines) would bind to virus and mediate effector functions even before the virus particle can enter a host cell. Such antibodies would be expected to be more effective than antibodies against secreted viral immuno-modulator – these are only produced once a host cell has been infected and virus replication must occur. It is important that the authors consider these differences and provide a balanced discussion.

RESPONSE: *The Discussion has been modified to indicate also that an immune response against virion proteins will be effective in protection (lines 531-532).*

The clinical use of TNFRs to treat various inflammatory conditions has shown that reactivation of a number of specific pathogens can occur in treated individuals. In their discussion, the authors should also discuss how combining a chemokine inhibitory domain to the TNFR therapeutics could potentially broaden the types of pathogens that may be reactivated or cause more serious disease.

RESPONSE: *The Discussion has been modify to mention that the anti-chemokine activity may increase the risk of infection in patients unders anti-TNF therapy (lines 544-547).*

Other comments:

Page 14, 3rd para, 3rd line – references not cited. Authors should be aware that lymphopenia in mousepox has been described several decades ago by Mims so those references should also be included and acknowledged.

RESPONSE: *The publications by Mims (C. A. Mims. Aspects of the pathogenesis of virus diseases. Bacteriol. Rev. 28, 30-71, 1964) has been included (line 335-336).*

There are several typographical errors throughout the manuscript.

RESPONSE: *We have corrected the typographical errors.*

REVIEWERS' COMMENTS:

Reviewer #1 (Remarks to the Author):

I have thoroughly read the revised manuscript, and all my concerns have been adequately addressed. For my part, the manuscript is ready for publication.

Reviewer #3 (Remarks to the Author):

The Authors have added new data or clarified with new explanations the issues I raised in my review. From my perspective, the incorporation of these changes has reinforced the conclusions of the study, which provides very interesting insights into virus-host interactions that extend beyond the biology of ECTV and poxviruses and can be of interest for a wide virology/immunology audience. I recommend publication.

Reviewer #4 (Remarks to the Author):

The authors have adequately addressed all of my concerns and comments. One thing that the authors should do is to provide the magnification for histology/immunohistochemical sections, including those of any insets in Figure 3 (panel c-h), Figure 5 (panel l-n) and Supplementary Figures 1 and 2.

NCOMMS-17-26267. Chemokines cooperate with TNF to provide protective anti-viral immunity and to enhance inflammation

Response to Reviewers' and Editor's comments

REVIEWERS' COMMENTS:

Reviewer #1 (Remarks to the Author):

I have thoroughly read the revised manuscript, and all my concerns have been adequately addressed. For my part, the manuscript is ready for publication.

Reviewer #3 (Remarks to the Author):

The Authors have added new data or clarified with new explanations the issues I raised in my review. From my perspective, the incorporation of these changes has reinforced the conclusions of the study, which provides very interesting insights into virus-host interactions that extend beyond the biology of ECTV and poxviruses and can be of interest for a wide virology/immunology audience. I recommend publication.

Reviewer #4 (Remarks to the Author):

The authors have adequately addressed all of my concerns and comments. One thing that the authors should do is to provide the magnification for histology/immunohistochemical sections, including those of any insets in Figure 3 (panel c-h), Figure 5 (panel l-n) and Supplementary Figures 1 and 2.

Response: We added the bars indicating the magnification of the tissue sections shown in Fig- 3, Fig.5, Suppl. Fig.1 and Suppl. Fig.2.

Response to modifications of the manuscript suggested by the Editor:

- We have deleted 7 paper citations, and the reference list now has 74 citations.
- We added a new Suppl. Fig. 3 showing the gating strategy used in flow cytometry.
- We included all methodology in the Methods section of the manuscript.
- The statistical tests are now indicated in the Figure legends.
- We added the link to the web page where viral genomic sequences are available.

The response to other minor comments are indicated in the manuscript file where changes are marked.